# Mutant KRAS vaccine with dual checkpoint blockade in resected pancreatic cancer: a phase I trial

Amanda L. Huff[1,2,3,12], S. Daniel Haldar[1,2,3,4,12], Alexander A. Girgis[1,2,3], Hejia Henry Wang [1,2,3], Ludmila Danilova [1,2,3], Thatcher Heumann[1,2,3], Maureen Berg[3], Yuxuan Wang[3], Lalitya Andaloori[1,2,3], Alexei Hernandez [1,2,3], Gabriella Longway[1,2,3], Benjamin Barrett[1,2,3], Zirui Zhu[1,2,3], Emily Davis-Marcisak[1,2,3,5], Christopher Thoburn [1,2,3], James Leatherman [1,2,3], Sarah Mitchell[1,2,3], Jae W. Lee[1,2,3,6], Daniel H. Shu[7], Maximillian F. Konig [3,8,9,10,11], Brian J. Mog[3,8,9,10,11], Janelle Montagne [1,2,3], Erin M. Coyne [1,2,3], Katherine Bever[1,2,3], Marina Baretti[1,2,3], Mark Yarchoan [1,2,3], Robert A. Anders[1,2,3], Luciane T. Kagohara [1,2,3], Daniel Laheru [1,2,3], Amy M. Thomas[3], Jennifer Durham[1,2,3], Julie M. Nauroth[1,2,3], Jiayun Lu[1,2,3], Hao Wang [1,2,3], Elana J. Fertig[7], Won Jin Ho [1,2,3], Nilofer S. Azad[1,2,3], Elizabeth M. Jaffee [1,2,3] ✉ & Neeha Zaidi [1,2,3] ✉

In this phase I study, we test a pooled synthetic long peptide vaccine targeting the six KRAS mutations (G12V, G12A, G12R, G12C, G12D, G13D) with ipilimumab and nivolumab in resected pancreatic adenocarcinoma. Co-primary endpoints include safety and maximal percent change of IFNγ-producing mutant KRAS T cell responses in the blood within 17 weeks. Secondary endpoints include disease-free survival, overall survival, and maximal percent change of IFNγ-producing mutant KRAS T cell responses at any time after vaccination. Vaccine-related adverse events are grade 1-2. 11/12 and 10/12 patients generate a significant increase in average T cell response to 6 mutant KRAS antigens and tumor-specific response, respectively. Immunophenotyping demonstrate Th1 CD4 central memory and effector memory T cells, and CD8 effector memory T cells at a lower frequency. The vaccine also generates cross-reactive T cells that recognize more than one mutant KRAS antigen. These findings support the safety and diverse anti-tumor immunity of mutant KRAS vaccines (NCT04117087).

Pancreatic ductal adenocarcinoma (PDAC) has a dismal 5–year survival of 12%[1]. The poor prognosis is due to the lack of early detection, early micrometastatic spread, and resistance to chemotherapy. Given that the majority of patients present with advanced disease, only 20% are eligible for surgical resection at the time of diagnosis[2]. Even after surgery and adjuvant chemotherapy, up to 80% of patients develop disease recurrence leading to poor long–term survival[3,4]. Immune checkpoint inhibitors (ICIs) alone are not effective in PDAC mainly due to a low tumor mutational burden, lack of functional effector T cells, and a highly immunosuppressive tumor microenvironment (TME)[5,6].

Oncogenic mutations in *KRAS* (mKRAS) are attractive targets due to their high prevalence in up to 90% of PDACs as well as, importantly, in precancer lesions[7,8]. As a driver mutation that is expressed clonally, mKRAS antigen loss is unlikely to cause tumor escape. Small molecule

mKRAS inhibitors have shown promise in advanced PDAC, but their use in the micrometastatic setting is unclear, and durability is limited by secondary resistance[9]. mKRAS has also been targeted with adoptive T cell therapies, but clinical benefit is restricted to patients with the few HLA subtypes in whom cognate T cells have been identified[10,11]. In contrast, mKRAS–targeted vaccines have the potential to broadly cover more HLA subtypes. Vaccines targeting mKRAS have been shown to slow tumorigenesis with a survival benefit in murine models of PDAC and lung cancer[12,13]. Prior clinical trials have tested mKRAS-targeted peptide vaccines[14-16] or peptide-pulsed antigen-presenting cells[17,18] but observed low immunological response rates (less than 50%) or were limited to single patient analyses. A more recent study using a lymph–node–trafficking mKRAS peptide vaccine showed positive peripheral immune responses but was limited to KRAS G12D and G12R[19]. To this point, little is known about the relative immunogenicity of the diverse mKRAS subtypes in broad patient populations. Furthermore, none of these studies have combined a pooled mKRAS peptide vaccine with ICIs that are known to reinvigorate exhausted T cells. Lastly, despite being the most common oncogene in human cancers, to our knowledge, -25 mKRAS-specific T cell receptors across the antigen subtypes have been sequenced and validated to-date, limiting the fields understanding of their cognate T cell recognition.

Here, we report the development of a pooled peptide vaccine––termed mKRAS-VAX––targeting the six most common KRAS mutations, KRAS G12D, G12V, G12R, G12C, G12A, and G13D. This vaccine provides the broadest coverage against mKRAS in a vaccine yet, which is important for potentially heterogenous expression of mKRAS in micrometastatic lesions and critical for use in the preventative vaccination setting[20]. In a first-in-human phase I clinical trial (ClinicalTrials.gov: NCT04117087), we tested mKRAS-VAX in combination with nivolumab (anti–PD-1 antibody) and ipilimumab (anti–CTLA–4 antibody) in patients with resected PDAC. Our trial was designed to determine safety and immunogenicity of mKRAS-VAX in combination with ICIs. We find that this combinatorial therapy is safe, feasible, and effective in inducing mKRAS–specific T cells across all six mKRAS antigens. High-dimensional immune profiling revealed activated and polyfunctional mKRAS-specific CD4 and CD8 T cells with distinct phenotypic profiles. Through bulk and single cell TCR sequencing, we generated the largest known database of putative mKRAS-specific TCRs from the peripheral blood of the vaccinated patients. Furthermore, we validated a subset of these TCRs functionally that respond to a single or multiple mKRAS antigens. Finally, we identified a public mKRAS TCR in more than one patient. The safety, immunogenicity, and broad coverage of mKRAS-VAX have provided the foundation for vaccinating high-risk individuals (ClinicalTrials.gov: NCT05013216)[21].

## Results

### mKRAS-VAX plus ICIs is safe and activates mKRAS-specific T cells

We enrolled and treated 12 patients (May 2020 to May 2023) with mKRAS-VAX, formulated with adjuvant poly-ICLC (Hiltonol®), in combination with ipilimumab and nivolumab (Fig. 1a, b, Supplementary Fig. 1a). Patients were eligible if they had undergone curative-intent surgical resection and standard-of-care adjuvant therapy within 6 months. Resected tumors were confirmed to have one of the six KRAS mutations. KRAS G12V was the most common mutation ($n = 6$), followed by G12D ($n = 4$) and G12R ($n = 2$) (Supplementary Table 1). Perioperative chemotherapy was administered to 11/12 patients. Median time from the completion of standard-of-care therapy to first vaccination was 4.2 months (range: 1.6-6.0). Importantly, time to first vaccination was dependent on when the patient was referred by their treating oncologist after completion of standard of care therapy. Additional baseline characteristics are included in Supplementary Table 1.

We achieved an acceptable safety profile of the combinatorial therapy, with the most frequent vaccine-related AEs being pain (83.3%), erythema (33.3%), muscle cramps (16.7%) at the injection site, fatigue (33.3%), and fever (33.3%) (Supplementary Table 2). These vaccine-attributable AEs were low-grade (grade 1-2) without grade 3-5 AEs. As expected from dual checkpoint blockade, there were four grade 3 immune-related AEs, namely pneumonitis, myalgias, arthralgias, and adrenal insufficiency. These events occurred in 2/12 patients (16.7%), resulting in discontinuation of ICIs, while booster doses of vaccine were continued (Supplementary Table 3,4).

We measured the magnitude of mKRAS-specific T cells in pre- and post-vaccine peripheral blood mononuclear cells (PBMCs) as the primary endpoint. Ex vivo restimulation with the six individual 21-mer synthetic mKRAS peptides or a control irrelevant fusion peptide was followed 24 h later by IFNγ ELISPOT (Fig. 1c, **Statistics in** Supplementary Table 5). We detected mKRAS-specific T cells in post-vaccine PBMCs but not in pre-vaccination samples. By determining the maximal fold change in response to each mKRAS antigen relative to baseline (Supplementary Fig. 1b), we found a statistically significant increase in average mKRAS-specific T cell response across all six antigens in 11/12 patients (91.7%); these patients were classified as 'immune responders' per the pre-specified primary endpoint (Fig. 2a). Furthermore, we detected a significant response to all six antigens within 17 weeks post-vaccination (Fig. 2b). Notably, the magnitude of T cell activation to each mKRAS antigen varied, possibly due to diverse HLA subtypes within the cohort (Supplementary Table 6). 7/12 patients (58.3%) mounted a positive response to all six mKRAS antigens with 11/12 (91.6%) responding to at least three antigens (Fig. 2c). 10/12 patients (83.3%) patients mounted a significant response to their tumor KRAS mutation, while 2 patients with a G12D mutation did not (Fig. 2d). mKRAS G12V and G12R peptides induced significantly higher levels of IFNγ-producing T cells relative to mKRAS G12C, G12D, or G13D (Fig. 2e). Lastly, we observed that the response to mKRAS G13D was associated with positive responses to G12D and G12R (Supplementary Fig. 1c).

With a median follow up of 35.8 months (data cutoff 10/24/2024), 4/12 patients (33%) remained disease-free, and 8/12 (66.7%) had disease recurrence. Four of the five patients who remained disease-free at the end of primary treatment phase entered the optional extended phase of the trial at 52 weeks, during which they received vaccine only (Fig. 1a). Patients with disease recurrence and those who remained disease-free had similar characteristics, including time from completion of standard adjuvant chemotherapy to first vaccination on study (Supplementary Table 1). Median disease-free survival (DFS) and overall survival (OS) of the entire cohort were 6.35 months and 29.59 months, respectively, as defined from the time of first mKRAS-VAX dose (Supplementary Fig. 2a,b).

Notwithstanding the small sample size, we elected to perform an exploratory correlative analysis between mKRAS-specific T cell responses and clinical outcomes. Patients were stratified into two groups above and below the 25th percentile either of the maximal fold change of the pooled average or the tumor mutation-specific T cell responses. Using either approach, patients within the upper three quartiles of T cell response displayed longer DFS than patients within the lowest quartile (median 18.8 vs 2.76 months, $P = 0.024$, hazard ratio (HR) = 0.19 (95% CI: 0.04−0.95) (Supplementary Fig. 2c). This analysis provides initial support for the hypothesis that strong responses to mKRAS are clinically meaningful in the context of resected PDAC, consistent with other recent study of adjuvant vaccines in PDAC[19,22]. There was no correlation between DFS and lymph node status, primary tumor size, baseline CA19-9 levels, ctDNA positivity at baseline, neoadjuvant chemotherapy, splenectomy, or absolute lymphocyte counts at baseline (Supplementary Fig. 2e-k). Of note is that all patients with a mKRAS G12D mutation recurred; 2 out of 4 were in the lowest quartile of average mKRAS immune responders (Supplementary

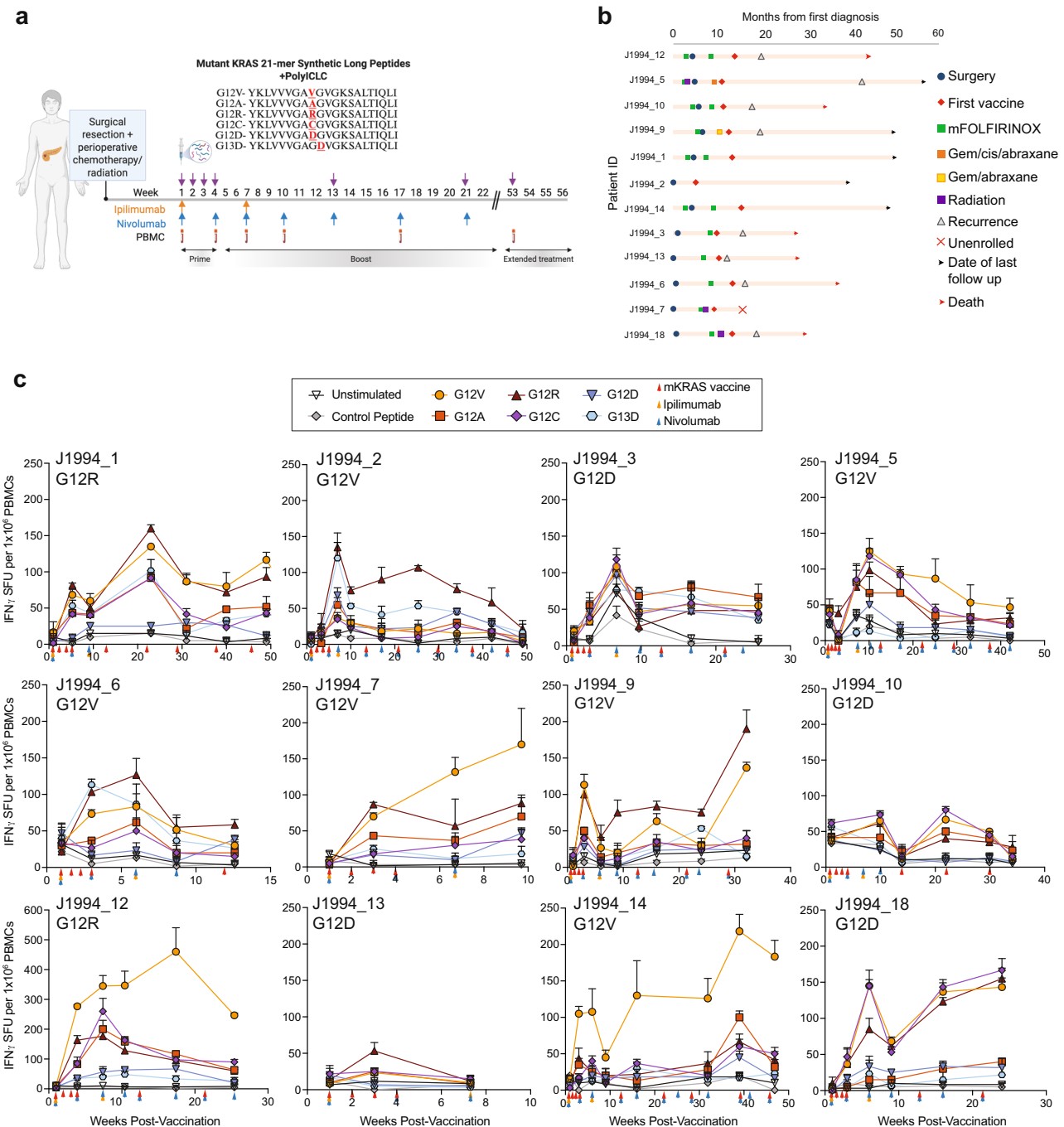

**Fig. 1 | A pooled mutant KRAS peptide vaccine given in combination with ipilimumab and nivolumab activates mKRAS-specific T cells in patients with resected PDAC. a** Treatment schema for patients treated with mKRAS-VAX, a pool of 6 synthetic long mKRAS peptides in the adjuvant setting. Patients receive four doses of 0.3 mg peptide/dose with 5ug polyICLC during the priming phase followed by booster vaccines every 8 weeks. Ipilimumab (1 mg/kg) and Nivolumab (3 mg/kg) were given for the first four doses followed by Nivolumab (480 mg) maintenance doses every four weeks. Created in BioRender. Huff, A. (https://BioRender.com/1vglal1) (**b**) Clinical event timeline for each patient enrolled and vaccinated. **c** IFNγ ELISPOT assay of PMBCs collected at baseline and post-vaccination time points after restimulation with vehicle only control (unstimulated) or 2ug/mL control peptide or individual mKRAS long peptides overnight. Statistics of IFNγ spot forming units detected after each mKRAS peptide stimulation relative to control peptide are shown in Supplementary Table 5. Data are shown as mean (*n* = 3) and upper limit of SD.

Fig. 2d). Those with a mKRAS G12D mutation also had a lower DFS compared with those with mKRAS G12V; this is consistent with the differences we noted in the immunogenicity for the respective antigens (Fig. 2e, Supplementary Fig. 2l). All four patients who remained disease-free at the time of data cutoff had mKRAS G12V or G12R tumors (Supplementary Fig. 2d). Finally, to determine if there was an association between baseline immunopathological features and DFS, we performed immunohistochemistry for CD4, CD8, MHC class II (HLA-

DR, DP, DQ), and PD-L1 on the pre-treatment, primary resected tumors (Supplementary Fig. 3a). There was no correlation between DFS and the densities or proportions of CD4, CD8 T cells, macrophages (CD68 + ), or PD-L1 expression within the primary tumor (Supplementary Fig. 3b-f). However, patients with moderate levels of HLA-DR, DP, DQ on tumor cells had a significantly higher DFS relative to patients with high or low/negative expression (Supplementary Fig. 3g).

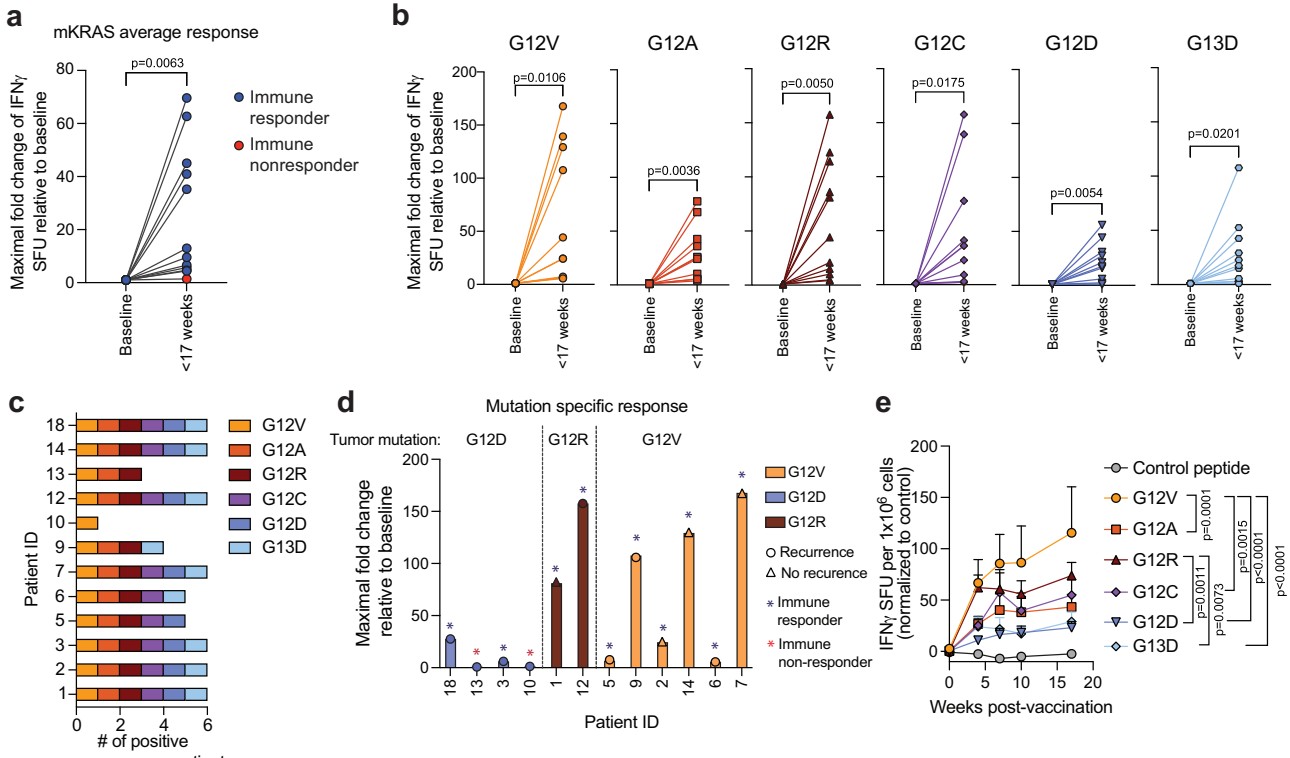

**Fig. 2 | Patients vaccinated with mKRAS-VAX mount T cell responses to mKRAS antigens with differential immunogenicity profiles. a** IFNγ response to each mKRAS antigen was calculated first relative to the control peptide stimulation for each mKRAS antigen at each time point within 17 weeks post first vaccine dose and then normalized to the baseline fold change. The maximal fold change reached for each antigen relative to baseline was then averaged for each patient designated as an immune responder (*n* = 11). Paired t test (two-tailed) was performed to calculate significance. Immune responders were defined as post-treatment responses over 2.77 times the standard deviation of the baseline. **b** Maximal fold change determined within 17 weeks for each mutation relative to baseline for immune

responders (*n* = 11). Paired t test (two-tailed) was performed to calculate significance. **c** Per patient, the number of mKRAS antigens in which a significant increase in IFNγ producing T cells after peptide stimulation were detected postvaccine relative to baseline (**d**) Maximal fold change response to each patient's tumor-matched mutation. **e** Magnitude of IFNγ producing mKRAS-specific T cells after restimulation with each mKRAS antigen or a control peptide across postvaccine time points for patients indicated as immune responders to the average mKRAS pool (*n* = 11). Mean and upper limit of SD are shown. Two-way ANOVA followed by Tukey's multiple corrections was performed.

## mKRAS-VAX plus ICIs activates polyfunctional Th1 CD4 and cytotoxic CD8 T cells

To phenotype the mKRAS-specific T cells, patient PBMCs were restimulated with individual mKRAS peptides followed by mass cytometry time of flight (CyTOF) using markers for T cell activation, effector function, exhaustion, and memory with analysis at pre-treatment (baseline), peak response (by IFNγ ELISPOT), and last collection (within the first 52 weeks) timepoints. Using unsupervised clustering, we identified CD4 and CD8 T cells within the total PBMC population, and activated populations enriched in mKRAS peptide restimulated PBMCs relative to a control peptide (Supplementary Fig. 4a-f). We next performed a refined clustering on both CD4 or CD8 T cells and identified two activated (Acv) mKRAS-responsive CD4 T cell populations and two Acv CD8 T cells populations (Fig. 3a). For all six neoantigens across all patients, mKRAS-specific CD4 T cells were the predominant population with responding CD8 T cells detected in a small proportion of patients for each antigen (Fig. 3b).

We quantified the percent of activated T cells relative to the control peptide stimulation at each timepoint. Within the CD4 compartment, we noted a significant increase in activated mKRAS-specific CD4 central memory (Tcm, CD45RO + CD45RA + CCR7 + ) and effector memory (Tem, CD45RO + , CD45RA-, CCR7-) populations at the peak and late timepoints relative to baseline (Fig. 3c). Notably, G12D and G13D had the lowest response magnitude, consistent with the ELISPOT data. The CD4 Tcm active population (Tcm Acv) was KI67+ and expressed the activation markers OX40, CD137, and CD25, with lower

levels of Th1 effector cytokines (IL-2, IFNγ, TNFα) and exhaustion markers (CTLA4, PD1, TIM3, and LAG3). Conversely, CD4 Tem Acv cells expressed the highest levels of activation markers (OX40, CD25, and CD137) and cytokines (IL-2 and TNFα) but also displayed a higher exhaustion marker expression (CTLA4hi, PD1hi, TIM3hi, and LAG3hi) (Fig. 3a). Both CD4 Acv populations expressed low levels of Granzyme B. We found no increase in unstimulated PBMCs in either activated or resting CD4 or CD8 T cell populations in post-vaccine samples validating our identification of peptide-specific, activated T cells (Supplementary Fig. 4g,h). We further confirmed the predominance of Th1 cytokines by Luminex multiplex cytokine secretion assay of supernatants from the peptide restimulated PBMCs (Supplementary Fig. 5). We detected lower secretion of the Th2-associated cytokine IL-5 but not of IL-4. We also detected the Th17-associated cytokine IL-17A after restimulation with certain peptides relative to control or unstimulated samples (Supplementary Fig. 5).

Within the CD8 compartment, the activated effector population (Teff Acv, CD45ROhi, CD28hi) expressed highest levels of the activation markers CD137, OX40, KI67, cytokines IFNγ, IL-2, and granzyme B, as well as the exhaustion markers CTLA4, TIM3, and PD1. This Teff Acv population was detectable in a limited number of patients for single antigen stimulation and was not significant across the cohort (Fig. 3d, right). Thus, this population, which exists at a relatively low frequency, represents highly activated but possibly more exhausted CD8 T cells. Conversely, a significant increase in activated CD8 effector memory cells (Tem Acv, CD45RAlo, CD45ROlo, CCR7-) was detected post-

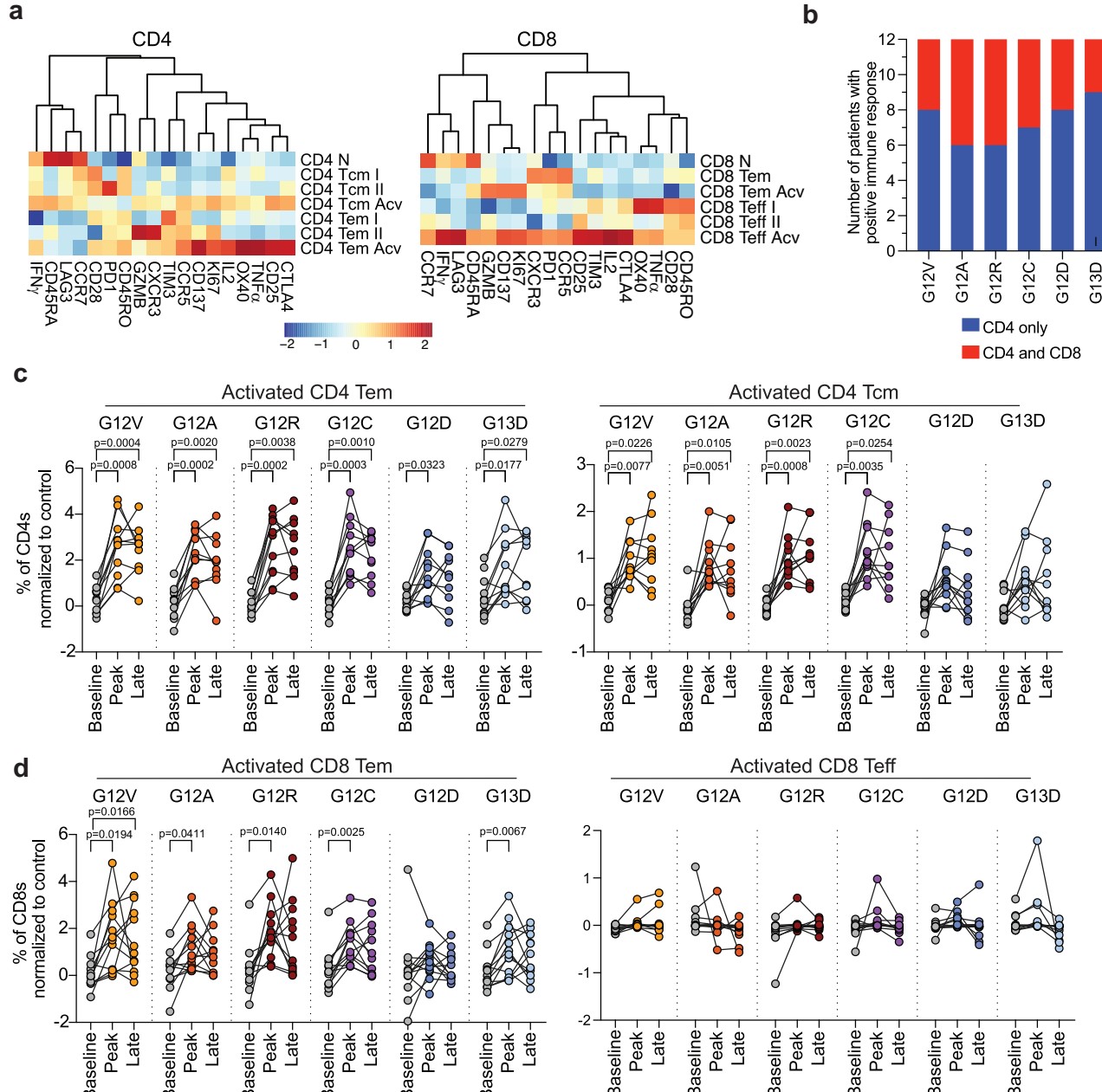

**Fig. 3 | mKRAS-specific T cells activated by mKRAS-VAX are polyfunctional Th1 CD4 and cytotoxic CD8 T cells.** PBMCs from all patients at three timepoints (baseline, peak of response measured by IFNγ ELISspot, and the last timepoint on trial with prime or booster phase) were restimulated with 2ug/mL of control peptide or individual mKRAS peptides for 48 h. Cells were then collected and stained with a panel of antibodies to determine T cell memory phenotype, effector function, and exhaustion profile and analyzed by CyTOF. **a** Unsupervised clustering of CD4 and CD8 T cell populations including CD8 effector (CD8 Teff) populations, CD8 effector memory (CD8 Tem), CD8 naive (CD8 N), CD4 effector memory (CD4 Tem), CD4 central memory (CD4 Tcm) and CD4 naive (CD4 N). Activated CD4 and CD8 T cell subtypes are indicated by Acv. **b** For each mKRAS antigen, the number of

patients with a positive CD4 only or CD4 and CD8 T cell response measured by cyTOF. **c** Percent of each activated population of total CD4s after peptide restimulation with individual mKRAS antigens relative to control peptide for CD4 T effector memory (Tem- left) or central memory (Tcm- right) activated or (**d**) Percent of each activated population of total CD8 T cells for activated CD8 effector memory (Tem- left) or effector activated (Teff- right). Each dot represents the magnitude of response to each antigen for each patient normalized to response to control peptide stimulation (Baseline-*n* = 12, Peak-*n* = 12, Late-*n* = 11). Mixed-effects analysis followed by Sidaks's multiple comparisons test for significance of peak and late time-point response relative to baseline sample stimulation of each corresponding antigen.

vaccine for G12V, G12A, G12R, G12C, and G13D across patients. This population expressed CD137 and KI67, with higher granzyme B expression and lower levels of dysfunctional markers (PD1, CTLA4, and TIM3) (Fig. 3a, d). Flow cytometry of peptide-restimulated PBMCs revealed an increase in CD69 and CD137 activation markers and effector cytokines (Supplementary Fig. 6a-e). Using an antigen stimulation titration assay, we also confirmed a lack of responsiveness of the

CD3+ compartment to wild-type KRAS relative to KRAS G12V (Supplementary Fig. 6f). Notably, in our CyTOF analysis, we detected the chemokine receptor CCR5 on both CD8 and CD4 mKRAS-responsive T cells, which has been associated with durable immunological memory and long-term therapeutic benefit in the context of ICIs (Fig. 3a)[23].

The long-term durability of mKRAS-specific T cells was assessed in all four patients who had remained disease-free at the end of the

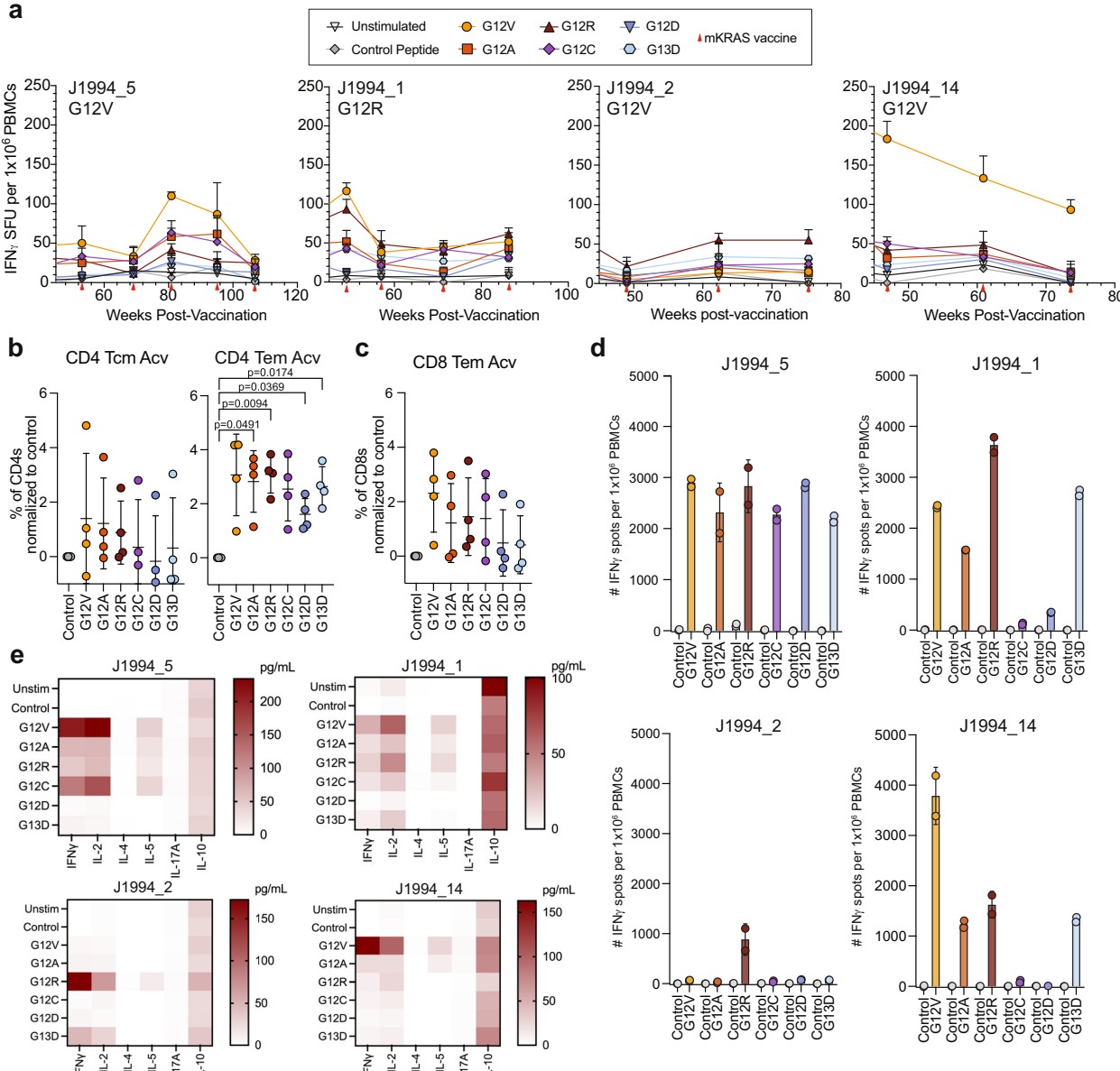

**Fig. 4 | mKRAS-specific Th1 CD4 T cells are detectable in patients in the extended phase of the vaccination schema.** Four patients were within the extended phase of treatment (>52 post first vaccine). **a** PBMCs at time points post 52 weeks from first vaccine dose were restimulated with vehicle only (unstimulated) or 2ug/mL control peptide or individual mKRAS peptides in anti-human IFNγ capture plates, overnight. The next days, cells were washed away, and the plate was analyzed for IFNγ spot forming units. Mean and upper limit of SD are shown. **b** CyTOF analysis of PBMCs restimulated with 2ug/mL control peptide or individual mKRAS peptides for 48 h. Percent of CD4 T central memory (Tcm- left) and effector memory (Tem- right) activated populations or (**c**) CD8 T effector memory (CD8

Tem) activated cells identified in the heatmap clustering in Fig. 3 (n = 4). Mean +-SD are shown. One-way ANOVA was performed followed by Dunnett's multiple comparisons test for significance relative to control peptide. **d** In vitro peptide expansion of PBMCs with individual mKRAS peptides of patients in extended phase of trial (J1994_5- week 106, J1994_1- week 106, J1994_2- week 126, J1994_14- week 106. Samples were ran as technical duplicates. **e** Luminex cytokine detection of IFNγ, IL-2, IL-4, IL-5, IL-17A, or IL-10 in the supernatant of control or peptide-restimulated PBMCs from each patient's last blood collection timepoint. Samples were ran as technical duplicates. One-way ANOVA was performed followed by Dunnett's multiple comparison test.

primary treatment phase and opted to receive booster doses of mKRAS-VAX alone every three months in an extended treatment phase (Fig. 4a). Measurable mKRAS-specific T cell ELISPOT responses, *albeit* of a lower magnitude, persisted in the extended phase (Fig. 4a, Statistics in Supplementary Table 7). Importantly, CyTOF of peptide-restimulated PBMCs detected the persistence of the highly activated CD4 Tem population, and to a more variable extent, the CD4 Tcm and CD8 Tem populations (Fig. 4b,c). In vitro peptide expansion of PBMCs from the last timepoint for each patient (>week 52) showed rapid expansion of mKRAS-specific T cells for all four patients (Fig. 4d). Consistent with earlier timepoints, supernatants from peptide-

restimulated PBMCs revealed predominantly Th1 cytokines, with lower but detectable Th2 and Th17 cytokines (Fig. 4e). In all, our comprehensive T cell phenotyping revealed the activation of distinct Th1 CD4 (predominant) and CD8 central and effector memory T cell populations in response to combinatorial therapy with vaccine and ICIs, and persistence of distinct populations with vaccine only.

## Identification of mono-reactive, cross-reactive, and public mKRAS-specific T cell clonotypes

The identification of mKRAS-specific TCRs is critical to better understand the biological underpinnings of mKRAS immunological

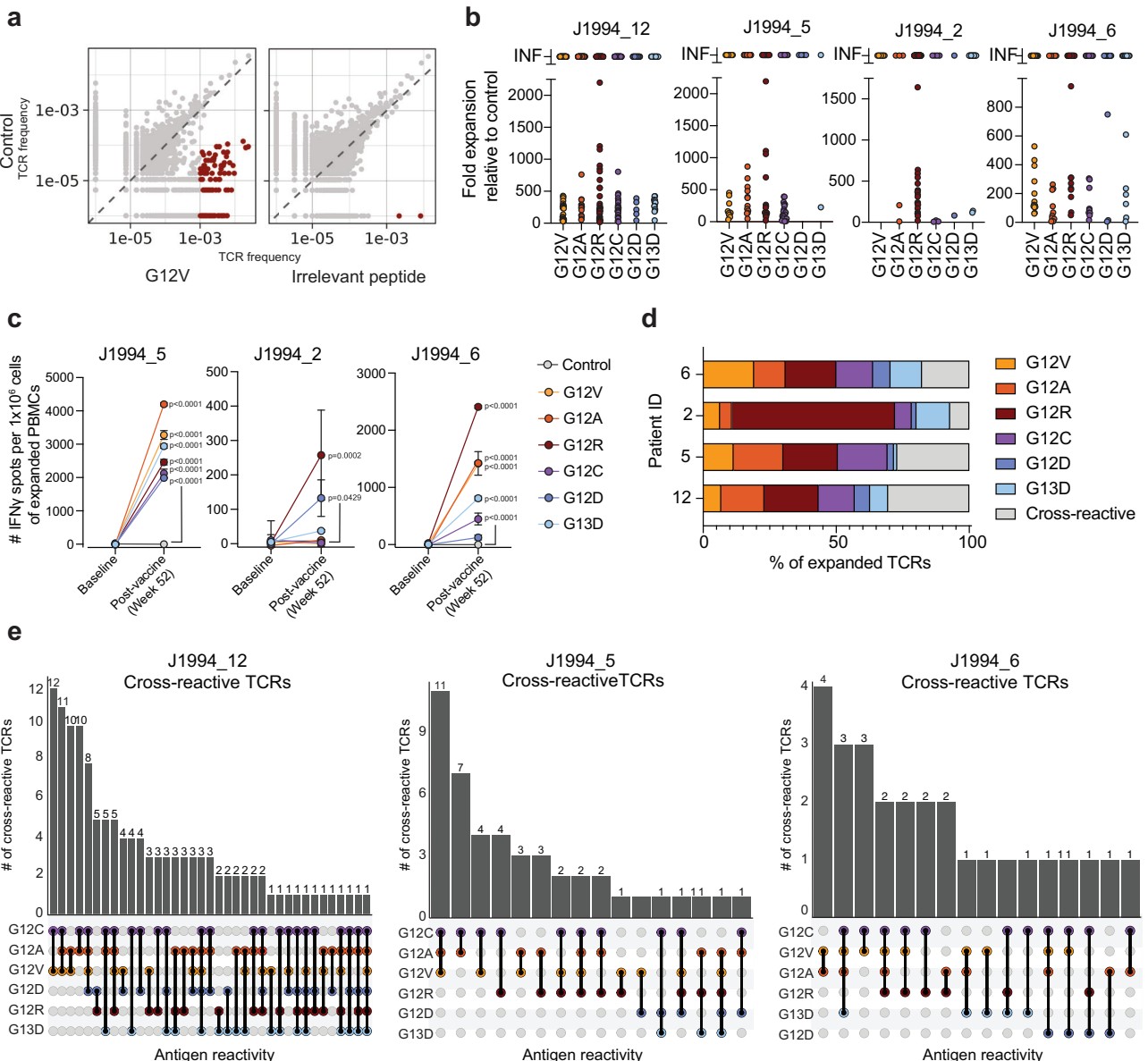

**Fig. 5 | Identification of mono- and cross-reactive mKRAS-T cells.** PMBCs from post-vaccine timepoints were incubated with 2ug/mL of irrelevant peptide or KRAS G12V, G12A, G12R, G12C, G12D, G13D peptide in the presence of 100IU/mL hIL-2 for 7 days. hIL-2 only was used as a control. Genomic DNA was isolated from each expansion and from a baseline sample and sent for TCRβ sequencing. Frequency of TCRs in the hIL-2 only control condition is shown relative to the frequency of each TCR in the peptide expansion conditions. **a** Representative plots of TCR frequency comparison for G12V expansion and irrelevant peptide expansion shown here. Significantly enriched TCRs are shown in red and number of significantly enriched is shown above each condition plot (Odds ratio >10, Frequency>0.1%). **b** Distribution of TCR clonotypes significant-ly expanded to each antigen or to

more than one antigen (cross reactive) in four patients. **c** Baseline or post-vaccine PBMCs underwent in vitro peptide expansion to indiviual mKRAS antigens followed by IFNγ ELISPOT to quantify mKRAS-specific T cells for 3 of the 4 patients with TCR sequencing data. Patient J1994_12 was not tested due to lack of baseline sample availability. Mean +-SD are shown. One-way ANOVA followed by Sidaks multiple comparison test was performed. **d** Distribution of TCR clonotypes significantly expanded to each antigen or to more than one antigen (cross reactive) in four patients. **e** For patients in which cross-reactive TCRs represented >10% of the expanded population (J1994_12, J1994_5, J1994_6), upset plots demonstrating number of TCRs that expanded to multiple mKRAS antigens.

recognition. Despite the predominance of the mKRAS oncogene in PDAC (up to 90%) and other cancer types (30%), few mKRAS-specific T cell clones have been identified to-date. One case report showed that an infusion of two CD8+ mKRAS-specific T cell clones recognized and killed lung metastases in a patient with a KRAS-expressing PDAC tumor, highlighting the importance of identifying other functional TCRs[11].

To isolate mKRAS-specific T cells, we performed in vitro peptide expansion with individual mKRAS peptides followed by bulk TCRβ sequencing in a subset of patients. To identify TCRβ sequences that expanded significantly in the presence of mKRAS peptides, we

compared the frequency of TCRβ sequences present in each mKRAS expansion condition to those in the irrelevant peptide or vehicle-only control (Fig. 5a). T cell clonal expansion was confirmed in response to mKRAS peptides relative to the unstimulated control for each patient as an increase in Shannon equitability, a measure of TCR repertoire clonality (Supplementary Fig. 7a). Based on recurrence status, we detected a trend, although not significant, towards more clonal repertoire of tumor-specific and average mKRAS-expanded TCRs in patients who did not recur (Supplementary Fig. 7b). Using stringent expansion cut-offs of an odds ratio (OR) of >5-fold relative to control and a frequency cutoff of >0.1% of the expanded repertoire, we

identified putative mKRAS TCRs for all four patients (Fig. 5b, Supplementary Fig. 7c, Supplementary Data 1). To confirm that mKRAS-specific T cells were not present in the pre-vaccine timepoint, we performed in vitro peptide-expansion assays with individual mKRAS peptides on PBMCs from the baseline sample followed by IFNγ ELISPOT for 3/4 patients (with sufficient baseline samples). No significant expansion of mKRAS-specific T cells was observed in baseline PBMC samples although significant expansion was detected in post-vaccine samples (Fig. 5c). Consistent with this, there was little overlap between the mKRAS-specific TCRs identified by expansion and bulk TCRβ sequencing of the resected primary tumors, despite the high numbers of productive TCRs sequenced for all tissue specimens (Supplementary Fig. 7d,e). Conversely, a single metastatic lung recurrence biopsied from patient J1994_5 had 80 unique mKRAS-specific TCR sequences overlapping within the tissue representing almost 20% of the mKRAS-repertoire identified for this patient (Supplementary Fig. 7e). Of the mKRAS-specific TCRs in the tumor, 55% were specific for the patient's tumor mutation (G12V) (Supplementary Fig. 7f). Thus, we reasoned TCRs present in an unstimulated baseline timepoint were unlikely to be mKRAS-specific and were filtered out of the expansion data; this further ensured the stringency of our inclusion criteria. Using this curated TCR repertoire, we observed that for 3 out of 4 patients, more than 10% of the enriched repertoire expanded to more than one mKRAS peptide, suggesting potential cross-reactivity of mKRAS-specific TCRs (Fig. 5d). For all three patients harboring cross-reactive TCRs, the most common antigen cross-reactivity was between mKRAS G12C, G12A, and G12V, and to a lesser extent, G12R (Fig. 5e).

We also identified TCRβ clonotypes that expanded in response to mKRAS peptides in multiple patients --termed 'public TCRβ clonotypes' (Supplementary Fig. 7g). Two public TCRβ clonotypes were also detected in a post-vaccine PBMC sample from a patient (J1994_8) with mKRAS positive metastatic colorectal cancer (CRC), who was also treated with mKRAS-VAX. These public TCRβ clonotypes expanded in response to at least one shared mKRAS antigen across patients, although some public TCRs expanded to different mKRAS antigens. Notably, public TCRβ06 (TCRBV28.01.01_CASRLGNTGELFF) was enriched in response to mKRAS G13D in two patients; this clonotype has been previously identified in tumor infiltrating lymphocytes as G13D-responsive in a patient harboring a G13D+ tumor[24]. In addition, public TCRβ06 and another public clonotype, TCRβ08 (TCRBV28.01.01_CASRRGNTGELFFF), which significantly expanded in response to mKRAS G13D, use the same TCRBV allele and differ in the antigen recognition CDR3 domain by a single amino acid, consistent with public "high similarity" (HS-TCRβ) clonotypes found to viral antigens. We asked if we could use HS-TCRβ comparisons to detect HS-mKRAS TCRβ clonotypes present in mKRAS positive PDAC tumor samples within the Cancer Genome Atlas (TCGA). Using our expanded TCRβ repertoire, we detected HS-TCR clonotypes with 1-2 amino acid differences, within the mKRAS-positive PDAC tumors, with 13 tumor samples having 10 or more HS-TCR clonotypes (Supplementary Fig. 7h, Supplementary Data 2). However, there was no correlation between the abundance of mKRAS HS-TCRs in a tumor sample and OS (Supplementary Fig. 7i). Overall, these data support the presence of public mKRAS-TCRβ clonotypes raised by mKRAS-VAX and imply the potential for these clonotypes to be present at baseline.

To determine the complete TCRαβ sequence of mKRAS-specific T cells in the four patients, we further performed scRNA/TCR sequencing on unstimulated PMBCs at pre- and post-vaccine timepoints. CD4 and CD8 T cell populations were identified based on cell-type marker expression (Fig. 6a, Supplementary Fig. 8a,b). No significant difference in post-vaccine TCR clonality was observed in patients who had recurred relative to those who had not recurred (Supplementary Fig. 8c). When looking at immune cells present pre- or post-vaccine, there was no significant change in the proportion of B cells, monocytes, dendritic cells, NK cells, or T cells across timepoints for all

patients (Supplementary Fig. 8d,e). At a gene level, we observed changes broadly in genes associated with metabolism, cytoskeletal structure, and inflammation in the post-vaccine CD4 Tcm and CD8 Tem populations, although we did not observe any significant changes at the pathway level by KEGG pathway analysis (Supplementary Fig. 8f, Supplementary Data 3). At a transcription factor level, we observed a trend towards increased *TBX21* (Tbet) but not *GATA3* or *RORC* (RORγt) expression in the CD4 Tcm subset post-vaccination (Supplementary Fig. 8g). Recently ZNF683, encoding the transcription factor hobit, has been shown to be a marker of a tissue-residency phenotype associated with expanding T cells in response to ICI[25]. We found that ZNF683 expression in proliferating CD8 T cells was significantly upregulated post-vaccine, potentially highlighting an ICI-induced population (Supplementary Fig. 8g).

To distinguish mKRAS-specific T cells in this dataset, we mapped mKRAS-specific TCRβ chains identified by in vitro expansion onto the scRNA/TCRseq dataset. mKRAS-specific TCRs mapped predominantly to CD4 Tcm (CD3 + CD4 + TNFRSF4 + TCF7 + IL7R +) and CD8 Tem (CD3 + CD8 + ITGB1 + CCL5 + CST7 +) populations, consistent with the populations identified by CyTOF (Fig. 6b, Supplementary Data 5). We noted that most CD8 T cell TCRs found in the adaptive sequencing datasets were present at a lower frequency ( < 0.1% of repertoire) compared to CD4 T cells; consistent with a predominant CD4 T cell response to our vaccine (Supplementary Fig. 8h). Therefore, to improve detection of potentially mKRAS-reactive CD8 T cells, we did not include a frequency cutoff and only considered TCRs that met the OR > 5 in mapping mKRAS-reactive T cells onto the single cell dataset (Supplementary Data 5). We compared the gene expression profiles of mKRAS-specific CD4 Tcm cells relative to all other CD4 Tcm cells in the dataset. We note that too few mKRAS-specific cells were detectable in the CD4 naïve, proliferating, or Tem populations to perform this analysis (Supplementary Fig. 8i, Supplementary Table 8). Five genes were significantly upregulated in the mKRAS-CD4 Tcm population, including *PDCD*, indicating higher activation/exhaustion levels; *GPR25*, whose homolog has been associated with T resident memory cells[26]; *APOBEC3H*, associated with inflammatory responses; *TRBV28* demonstrating a V gene enrichment for mKRAS-reactive CD4 T cells; and *TBX21* (Tbet), consistent with the trend towards increased expression in the broad CD4 Tcm population and the functional Th1 phenotype observed after antigen restimulation (Fig. 3). Similar expression levels of markers associated with central memory phenotypes and stem-like cell phenotypes, including *LEF1* and *TCF7*, and senescence and dysfunction-associated markers, including *FOXP3, KLRG1, LAG3*, and *CTLA4*, were observed between mKRAS-specific CD4 Tcm and all other CD4 Tcm cells (Supplementary Fig. 8i).

## Validation of antigen specificity of mutant KRAS TCRs

We sought to validate the antigen-specificity of the mKRAS CD4 and CD8 TCRs identified in the peptide expansion assay that were mapped onto the single cell dataset (Supplementary Data 5). For this, JurkatTCR_KO-NFAT-GFP cells were transduced with lentivirus expressing mono-reactive, cross-reactive, or public putative mKRAS-reactive TCRs in our dataset and were flow sorted for TCR+ cells (Supplementary Fig. 9a,b). TCR+ cells were co-cultured overnight with patient-matched B lymphoblastoid cell lines (LCLs) and pulsed with individual mKRAS peptides, control peptide, or wild-type KRAS synthetic long peptides. GFP expression was quantified by flow cytometry to assess antigen recognition (Supplementary Fig. 9c). All 4 mono-reactive TCRs responded to their respective mKRAS antigen that elicited expansion and did not respond to other mKRAS mutations or wild-type KRAS (Fig. 6c). Cross-reactive TCR05 responded to G12V, G12A, G12C, and G13D with weak reactivity to WT KRAS observed (Fig. 6c). TCR03, a cross-reactive TCR that had expanded in vitro to KRAS G12V, G12A, and G12C responded significantly to these three antigens at the highest peptide concentration tested (10 μM) but did not respond at lower

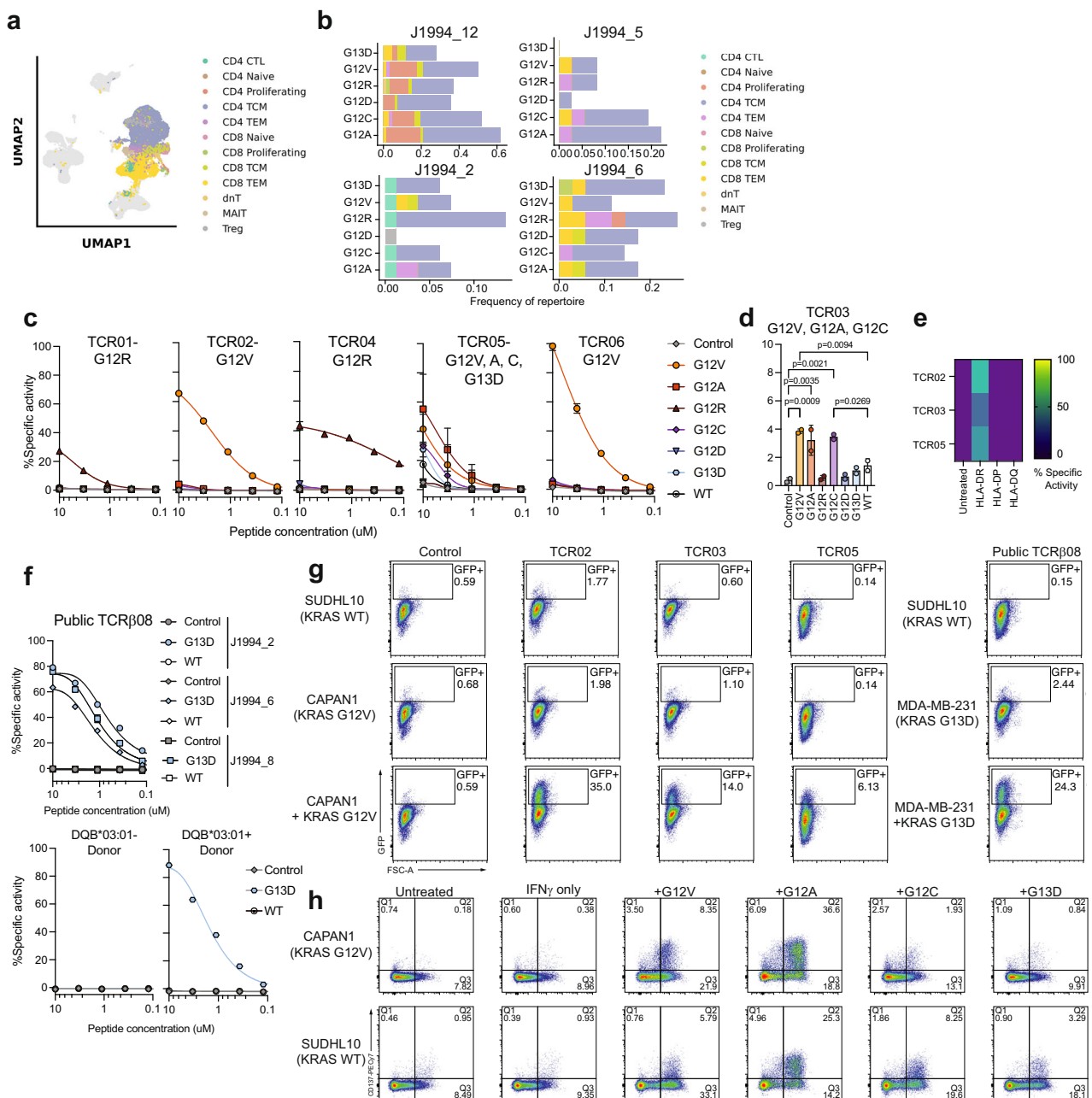

**Fig. 6 | Validation of mKRAS-specific T cell reactivity identified from peripheral blood of vaccinated patients. a** UMAP of single cell RNA/TCR sequencing performed on pre- and post-vaccine PBMCs. T cells were identified as CD4 (naïve, proliferating, central memory (TCM), and effector memory (TEM)), CD8 (naïve, proliferating, TCM, and TEM), double negative T (dnT), MAIT, and Tregs. **b** Phenotype of mKRAS-specific T cells identified from mKRAS expansion mapped onto single cell data by TCRβ chains (**c**) Specific activity of Jurkat-TCRKO-NFAT-GFP cells expressing putative mKRAS-specific TCRs and co-cultured with patient-matched LCLs pulsed with control peptide, mKRAS peptides, or KRAS WT peptide. Mean +-SD are shown. **d** Specific activity of Jurkat reporter line transduced with TCR03 and co-cultured with patient-matched LCLs pulsed with 10uM control, mKRAS peptide, or wild-type KRAS peptide. Mean +-SD are shown. One way ANOVA was performed followed by Tukeys multiple comparisons. **e** G12V reactive TCRs cocultured with HLA-null K562s transfected with patient-matched HLA class II

alleles and pulsed with 2ug/mL KRAS G12V peptide. Co-cultures were assessed for GFP expression. **f** Specific activity of Jurkat-Public TCRβ08 expressing cells co-cultured with patient matched LCLs pulsed with control, KRAS G13D, or wild type peptide at the indicated concentrations. Specific activity of public TCRβ08 Jurkats co-cultured with LCLs from a HLA-DQB*03:01- donor (bottom left) or irrelevant HLA-DQB*03:01+ donor (bottom right) pulsed with control, KRAS G13D, or wild type KRAS peptide. **g** Activation (GFP expression) of KRAS TCR+ jurkat cell lines after co-culture with human CAPAN1 (KRAS G12V), SUDHL10 (KRAS WT), or MDA-MB-231 (KRAS G13D) pretreated with 100IU hIFNγ for 48 h at an effector to target ratio of 2:1 for 24 h in the presence or absence of 2ug/mL exogenous mKRAS antigen. A negative control TCR recognizing a SARS-CoV2 was used. **h** CD25 and CD137 expression on KRAS TCR05 knock-in healthy donor T cells 24 h after co-cultured with IFNγ pre-treated CAPAN1 or SUDHL10 cells in the absence or addition of 2ug/mL additional exogenous mKRAS antigen.

concentrations of peptide (Fig. 6e). However, the avidity of this TCR for each mutation was notably weaker than the other TCRs. To determine the HLA-specificity for each TCR derived from patient J1994_2, HLA-null K562 cells were co-transfected with plasmids

expressing the HLA-DR, DP, or DQ alpha and beta chain alleles and co-cultured with TCR transduced Jurkat cells. All mKRAS-specific TCRs recognized cognate peptide in the context of the HLA-DR*07:01 allele (Fig. 6e). Notably, TCR05 was indicated from the single cell dataset as a

CD8 T cell; however, it responded to antigen in the context of this class II allele suggesting it is likely a CD4 T cell.

Additionally, we confirmed the antigen specificity and HLA-restriction of public TCRβ08, which was identified as significantly expanded to KRAS G13D in the bulk TCRβ sequencing for three patients (J1994_2, J1994_6, J1994_8) relative to control (Supplementary Fig. 7g). The paired alpha chain for this TCRβ sequence was identified only in the single cell dataset for patient J1994_2 and was therefore cloned using this alpha-beta chain pairing. Public TCRβ08 responded to KRAS G13D but not to control or wild-type KRAS peptide in the context of peptide pulsed LCLs for the three patients it was detected in (Fig. 6f). Between these three patients, HLA-DQB*03:01 was the only shared class II allele (Supplementary Table 6). To test if this allele was required for TCR recognition, we co-cultured Public TCRβ08 with peptide pulsed LCLs from a donor that did not have the DQB*03:01 allele. TCRβ08 failed to respond to G13D when co-cultured with this DQB*03:01 negative donor, while co-culture with LCLs from an irrelevant DQB*03:01 positive donor pulsed with G13D allowed for efficient TCR activation (Fig. 6f).

To determine if these mKRAS-specific T cells were capable of direct tumor recognition, we tested TCR02, TCR03, and TCR05 which recognize KRAS G12V in the context of HLA-DR*07:01 against the HLA-matched human pancreatic cancer line CAPAN1 which endogenously expresses KRAS G12V or the control line SUDHL10 expressing KRAS WT. We also tested recognition of public TCR08 (G13D-reactive) against target cell line MDA-MB-231 a KRAS G13D+ tumor line expressing HLA-DQB*03:01. We pre-treated the tumor cell lines with IFNγ for 48 h to upregulate HLA class I and class II on the surface (Supplementary Fig. 9d), followed by co-culture with TCR transduced Jurkat reporter lines for 24 h in the absence or presence of additional exogenous KRAS G12V peptide. High tumor recognition, measured by an increase in GFP expression from the reporter line after co-culture, was observed predominantly when exogenous G12V was supplied to the CAPAN1 co-cultures (Fig. 6g). This was consistent for public TCRβ08 when co-cultured with the KRAS G13D+ target cell line MD-MB-231 (Fig. 6g). None of the TCRs responded to the WT KRAS-expressing SUDHL10 control line, confirming a lack of recognition of healthy tissue. To determine the potential for clinical translation of these TCRs, and to test cross-reactivity of TCR05 against multiple tumor antigens presented on tumor cell lines, we generated a transgenic human TCR expressing line from healthy donor T cells in which endogenous TCR was knocked out followed by lentiviral overexpression of TCR05 (Supplementary Fig. 9e). After co-culture with KRAS G12V + CAPAN1 cells pre-treated with IFNγ, strong activation was detected only when each mKRAS antigen was supplied exogenously to either CAPAN1 or SUDHL10 cells, which expresses the cognate HLA allele (Fig. 6h). To test if TCR05 can indirectly recognize target antigen, we pulsed patient matched LCLs with tumor cell lysate generated from control or CAPAN1 (G12V +) tumor cells (Supplementary Fig. 9f). Indeed, TCR05 expressing human T cells increased CD25 expression after co-culture with CAPAN1 but not control lysate pulsed APCs supporting a role for indirect recognition of tumor with these identified mKRAS-specific TCRs.

## Discussion

Most patients with PDAC experience recurrence after surgery and chemotherapy, despite current optimal standard-of-care treatment, highlighting the need for novel strategies to eliminate micrometastatic disease. Immunotherapy has historically shown limited success in PDAC, but recent advances, including a bespoke neoantigen vaccine[22] and a mKRAS vaccine targeting two common KRAS mutations[19], have demonstrated initial evidence of clinical activity in patients with resected PDAC. In contrast to these studies, we evaluate a pooled synthetic long peptide vaccine targeting the six most common mKRAS subtypes in combination with two ICIs in resected PDAC patients. The

goal of this study was to conduct an in-depth analysis of differential T cell activation and expansion to each of the six KRAS mutations included in the vaccine. We show that a pooled KRAS vaccine covering the common KRAS mutations in PDAC administered in an HLA-agnostic manner is feasible and can induce robust tumor-specific immunity in 83% of patients. Through ex vivo peptide expansion and TCR sequencing, we describe a repertoire of 329 unique mKRAS-specific TCRs with reactivity to the six most common mKRAS antigens, expanding the field's database of mKRAS-reactive TCRs. Our data not only provides evidence for safety of our combinatorial approach, but also insights into T cell-specific immunogenicity of the most conserved oncogene expressed in human cancers.

Patients were allowed to enroll within 6 months of completing adjuvant chemotherapy and therefore joined the study at different timepoints within this window. Our exploratory analysis suggests that patients in the top three quartiles of average tumor-specific T cell responses (by ELISPOT) experienced greater DFS, which we have defined from the time of first vaccination. Notably, we did not observe a correlation of DFS with patient characteristics, tumor size, lymph node status at resection, CA-19-9 levels, ctDNA positivity, neoadjuvant chemotherapy, splenectomy, or lymphocyte count at baseline. Although perhaps informative for a potential magnitude of response that is required for clinical benefit, these correlations should be interpreted with caution given that the trial was not powered for clinical efficacy but appears to be consistent with recent neoantigen vaccine studies in PDAC[19,22]. Our study was also limited in analysis of post-vaccine tumor samples which were unavailable in this cohort. Future studies which correlate mKRAS-specific T cell infiltration into tumors with clinical outcomes may also be more informative.

Little is known about the immunogenicity ranking of antigens included in vaccines, particularly when administered as a pooled vaccine. Here, we are the first to test a pooled vaccine against 6 common KRAS mutations. Interestingly, we observed weaker immunogenicity of mKRAS G12D independent of whether the patient's tumor expressed this mutation. In contrast, T cell responses to mKRAS G12V and G12R were of a higher magnitude than mKRAS G12D and G13D, findings that are consistent with preclinical models[27]. As KRAS G12D is the most common mutation detected in PDAC tumors (~40%), this highlights the need to improve immunological responses against this mutation. One approach may be vaccination with a single tumor-specific peptide vaccine in the therapeutic setting, which could be explored in future studies. However, in the preventative vaccination setting where a potential KRAS mutation is unknown, single peptide vaccination limits immunological coverage across mutations. Additionally, another question remains if the cross-reactive T cells raised by the pooled vaccine in this cohort may also be advantageous for providing broader immunological protection across antigens. The low immunogenicity of mKRAS G12D also coincides with decreased DFS, suggesting that different mutations may have different rates and/or mechanisms of tumor growth and TME immunosuppression[28,29]. Reduced OS has been shown for mKRAS G12D positive tumors compared with G12R tumors, but not relative to G12V tumors. However, in our *albeit* 12 patient cohort, we observed a significant improvement in DFS for patients with G12V mutations relative to G12D positive patients[30]. Larger studies are required to further examine the clinical significance of this preliminary observation.

Cancer vaccines must not only generate a diverse activated T cell response for tumor killing but also elicit memory T cells should recurrence occur. We show, through single cell proteomics and transcriptomics, that mKRAS-VAX activates mKRAS-specific Th1 CD4 central and effector memory T cells that express functional cytokines IL2, IFNγ and TNFα as well as cytotoxic effector memory CD8 T cells which express the cytotoxic effector molecules GzmB. The Th1 CD4 responses observed are consistent with other peptide-based vaccine approaches and have previously been associated with improved

clinical outcomes[31]. Notably, and consistent with our approach, the amphiphile-peptide mKRAS vaccine also elicited predominantly Th1 CD4 T cells, while in contrast no mKRAS-specific T cells were detected by mRNA-based vaccine[19]. Additionally, the mRNA-based approach mounted predominantly CD8 T cells to the targeted personalized neoantigens, highlighting the need for more study of the differences of T cell populations activated by each platform[22]. Lower levels of the Th2-associated cytokine IL-5, but not IL-4, were also detected after mKRAS peptide stimulation in this study. While IL-5 function is still not entirely understood in the context of immunotherapy, recently, IL-5 secreting CD4 T cells have been associated with efficacy of ICB by promoting eosinophil expansion, tumor infiltration, and thus intra-tumoral CD8 T cell activation in breast cancer[32]. In our study, we found that the activated T cell populations persisted and tracked with a durability marker CCR5 in the four patients who remained enrolled during the extended vaccine-only, no-ICI phase. The CD4 Tem population was the most durable, likely owing to the trafficking of CD4 Tcm to the secondary lymphoid tissues over time due to CCR7 expression. Furthermore, granzyme B was detected in responding mKRAS-specific CD4 T cells, questioning the potential for a role of CD4 T cells in direct tumor killing. In the small subset of TCRs validated in our study, T cells recognized antigen when peptide was supplied exogenously or when KRAS-mutation specific tumor cell lysate was pulsed onto APCs, suggesting their function in controlling tumor growth may be more related to indirect mechanisms of tumor control as previously reported[33–37]. Notably, Alspach et al described the critical role of neoantigen-specific CD4 T cell-mediated control of tumor growth in tumors that lacked expression of MHCII, underscoring the potential importance of these T cell responses[33–37]. We also found that four patients who remained recurrence free displayed moderate levels of HLA-DR, DP and DQ in their primary tumor tissues, again suggesting that CD4 T cells may directly recognize patient tumors. However, additional testing of mKRAS-specific CD4 T cells in our dataset and their ability to lyse tumor directly needs to be explored. Nonetheless, these data appear to be consistent with high levels of HLA class II associated with greater activation of T regulatory cells and a more immunosuppressive TME[38].

We identified a repertoire of mono-reactive and cross-reactive mKRAS-specific T cells. Consistent with previous studies[19,39–41], KRAS G12V, G12C, and G12A were the most common cross-reactive targets. These three mKRAS antigens also displayed high immunogenicity rankings in our trial patients. Notably, the mono-reactive CD4 T cells generally had higher avidity for their target peptide-HLA complex relative to the cross-reactive TCRs; the cross-reactive TCR displayed low affinity to the cognate antigens G12V, G12A, and G12C. Furthermore, when the cross-reactive TCR05 was cloned into human T cells, it recognized four different mKRAS antigens but only displayed a low-level reactivity to wild-type KRAS. This patient and all others on the trial had no vaccine-related AEs, suggesting that a low level of wild-type cross-reactivity might be tolerable. Thus, while mKRAS-VAX generates a diversity of high and low avidity KRAS-reactive TCRs, these data bring into question the relevance of low avidity TCRs. Low avidity T cells are known to make up most of the primary immune response and remain persistent over time relative to high avidity T cells[42,43]. Further, low avidity T cells have been shown to display a lower exhaustion profile and may also be converted to high avidity T cells upon treatment with IC modulators, including a T cell agonist[44–46]. Further studies are required to explore the functional role of low versus high avidity mKRAS-specific T cells and their durability over time.

Importantly, we also identified public mKRAS TCRβ clonotypes within our dataset. One of these mKRAS TCRs identified in two patients on our trial was also detected in the tumor infiltrating lymphocyte population of a patient by Levin et al.[24]., further supporting the detection of public clonotypes across patients outside of our cohort. The alpha chains of the public TCRβ06 found in a previous study and

HS-public TCRβ08 found in our study were different; however, both clonotypes responded to KRAS G13D indicating a potential TCRβ chain-dominant interaction with cognate peptide-HLA[24]. We confirmed TCRb08 had conserved reactivity to the G13D antigen in the context of DQB*03:01 identifying that patients with this mutation-HLA combination may be an ideal population for vaccination with this antigen in future clinical settings. Additionally, these public clonotypes may be ideal candidates for adoptive TCR therapies based on their conserved immunological relevance across patients. While public TCR clonotypes have been identified against many pathogens[47], little is known about the abundance and role of public or HS-public clonotypes against cancer antigens. Thus, this work provides important basic immunological context of shared T cell responses to these highly conserved neoantigens across human solid tumors. To our knowledge, this is the first report of public mKRAS-specific TCR clonotype detected in patients vaccinated with any mKRAS vaccine.

In summary, we provide clear evidence for safety of our combinatorial immunotherapy with mKRAS-VAX plus ICIs. We also offer an in depth understanding into vaccine-ICI induced immunogenicity of the six common mKRAS antigens in PDAC. Specifically, we provide granular detail on vaccine-induced T cell phenotype and repertoire, as well as, importantly, mKRAS subtype differences in the T cell response. We believe that these insights are critical to the burgeoning field of cancer immunology in enabling the iterative design of future pooled mKRAS vaccines for optimal efficacy in PDAC and other mKRAS-driven cancers. As an off-the-shelf vaccine, this approach is also amenable to diverse patient populations without the need for HLA typing. This breadth also allows for its use in interception studies in high-risk individuals, such as the one underway (NCT05013216).

## Methods

### Study design, patient enrollment, and treatment schema

This study was conducted in accordance with the Declaration of Helsinki and good clinical practice guidelines. The trial protocol and amendments were approved by the Institutional Review Board at Johns Hopkins University and the US Food Drug Administration. This trial was registered on ClinicalTrials.gov (NCT04117087). All participants provided written informed consent. No financial incentive was provided. Participants were recruited from Johns Hopkins under IRB approved protocol. The study protocol is available in the Supplementary Information. Data collection was collected prospectively at the time of enrollment and during protocol-defined visits. Both male and female participants were included in this study. Sex and gender were recorded based on clinical documentation in the electronic record based on self-report. Sex was reported as a descriptive demographic variable. Analysis was not stratified on sex/gender because the study was not powered to detect sex-specific differences in outcomes. Demographic variables, disease characteristics, and treatment information were abstracted from the electronic medical record. Radiographic assessments were performed at pre-specified intervals. Laboratory and correlative studies were performed on prospectively collected biospecimens. All data were recorded in a secure, IRB-approved research database at Johns Hopkins.

The co-primary endpoints were safety and immunogenicity. Immunogenicity was defined as the maximal fold change in the pooled average mKRAS-specific T cell density across six mutation subtypes within 17 weeks post-vaccination using serial IFNγ ELISPOT assays of peripheral blood. Safety was assessed by the frequency and type of treatment-related adverse events per NCI Common Terminology Criteria for Adverse Events, version 5.0.

This single-arm, open-label, first-in-human phase I trial was conducted at the Johns Hopkins Sidney Kimmel Comprehensive Cancer Center to assess the safety and immunogenicity of a pooled synthetic long peptide mutant KRAS vaccine (mKRAS-VAX) combined with ipilimumab and nivolumab in patients with resected PDAC

(NCT04117087) with a start date of 5/28/2020. Patients were eligible to participate following curative-intent surgery and completion of adjuvant chemotherapy and/or radiation therapy per standard-of-care. Molecular tumor testing must have confirmed the presence of one of six KRAS mutations included in the study vaccine. KRAS testing was performed locally in a CLIA certified laboratory using next generation sequencing (NGS) or external CLIA-certified laboratories. Only those patients who harbor one of the KRAS mutations were eligible to enroll.

On-study treatment was divided into prime and boost phases. In the prime phase, patients received mKRAS-VAX on days 1, 8, 15, and 22 with two doses of ipilimumab 1 mg/kg and four doses of nivolumab 3 mg/kg. In the boost phase, patients received mKRAS-VAX every 8 weeks with nivolumab 480 mg every 4 weeks, up to a total of 14 treatment cycles. In an optional extended phase, patients who remained disease-free were permitted to receive four additional booster vaccines every 3 months during a second year on study. Radiographic disease assessments by CT imaging were performed every 3 months.

## Vaccine formulation and administration

The study vaccine consists of 21-mer synthetic long peptides corresponding to six of the most common KRAS mutation subtypes found in PDAC. Peptides were synthesized by JPT Technologies with sequences as follows: G12V- YKLVVVGA$\underline{V}$GVGKSALTIQLI, G12C- YKLVVVGA$\underline{C}$GVGKSALTIQLI, G12A- YKLVVVGA$\underline{A}$GVGKSALTIQLI, G12R- YKLVVVGA$\underline{R}$GVGKSALTIQLI, G12D- YKLVVVGA$\underline{D}$GVGKSALTIQLI, G13D- YKLVVVGAG$\underline{D}$GVGKSALTIQLI. These six peptides were admixed (0.3 mg/peptide, a total of 1.8 mg) with 0.5 mg TLR3 agonist poly-ICLC (Hiltonol; Oncovir) in saline prior to vaccine administration at the Investigational Drug Service Pharmacy at Johns Hopkins Hospital. The pooled vaccine consisting of all six peptides was administered subcutaneously at five injection sites. For in vitro assays, lyophilized peptides were dissolved in DMSO at 4 mg/ml and aliquots were stored at -80 °C. The sequence for the control peptide used in in vitro stimulations corresponds to an irrelevant fusion gene product as follows: RKREIFDRYGEEVKEFLAKAKEDF. If stability of peptides dropped below 90% or if sterility did not meet criteria, peptides were excluded from vaccine timepoints as indicated (Supplementary Table 9).

## Statistical analysis

Patients were considered evaluable for immunogenicity if they received at least one vaccination and had peripheral blood samples available at baseline and within 17 weeks post-vaccination. Descriptive statistics, boxplots, and plots over time were used to summarize mKRAS-specific T cell responses. Number of IFNγ spot forming unit (SFU) after peptide restimulation as a measure of T cell activation was normalized to an irrelevant control peptide stimulation at each time point. T cell measurements at each time point were reported using two approaches: 1) pooled average across six mKRAS subtypes and 2) response specific to the mKRAS subtype found in each patient's tumor. Fold-change from baseline is calculated at each time point, and T cell response is defined as maximum fold-change within 17 weeks. To determine whether the change in T cell is a true change for individual patient and not due to random variation, the difference of T cell IFNγ SFU between pre- and post-vaccination was compared to the threshold of 2.77 times standard deviation (SD) of the baseline values across patients. For each patient, the observed change greater than 2.77 SD is deemed a statistically significant change at the two-sided significance level of 0.05[48]. Disease-free survival and overall survival were evaluated using the Kaplan-Meier method. Statistical analysis was performed using Rstudio version 4.3.0 (Rstudio Inc). For each patient, the observed change greater than 2.77 SD is deemed a statistically significant change at the two-sided significance level of 0.05[48]. Disease-free survival and overall survival were evaluated using the Kaplan-Meier method. The log-rank test was used to compare survival times between patient subgroups, and hazard ratio was estimate based on

Cox model. Statistical analysis was performed using Rstudio version 4.3.0 (Rstudio Inc).

Genomics analyses and graphing were performed in R as described above, and graphing and statistical analysis for the experimental data was performed using GraphPad Prism 9 (Version 9.4.0). Multiple comparisons were analyzed using two-way ANOVA followed by Dunnett's multiple comparisons, Sidak's multiple comparisons, or Tukey's multiple comparisons test where appropriate. Data is presented as group mean ± standard deviation or ± standard error of the mean where indicated. Statistical significance was set as ns = $p > 0.05$, $*p \leq 0.05$, $**p < 0.005$, $***p \leq 0.0005$, $****p \leq 0.00001$. Figures were prepared using Adobe Illustrator CC 2022.

## ctDNA detection from plasma samples

ctDNA analysis was done in the Ludwig Center at Johns Hopkins. DNA was extracted from 5 mL of plasma per patient using cfPure MAX Cell-Free DNA Extraction Kit (BioChain, cat#K5011625MA), as specified by the manufacturer's instructions. DNA from peripheral white blood cells (WBCs) was purified with the QIAsymphony DSP DNA Midi Kit (Qiagen, cat#937255) according to manufacturer's instructions. DNA was then made into SaferSeqS libraries and queried for common driver mutations[49]. ctDNA analysis was done in the Ludwig Center at Johns Hopkins. DNA was extracted from 5 mL of plasma per patient using cfPure MAX Cell-Free DNA Extraction Kit (BioChain, cat#K5011625MA), as specified by the manufacturer's instructions. DNA from peripheral white blood cells (WBCs) was purified with the QIAsymphony DSP DNA Midi Kit (Qiagen, cat#937255) according to manufacturer's instructions. DNA was then made into SaferSeqS libraries and queried for common driver mutations **in genes including** *TP53, KRAS, CDKN2A, SMAD4,* and *APC* (ref. 49). Any mutation that was present in the matched WBCs from the patient was attributed to clonal hematopoiesis of indeterminate potential (CHIP) and therefore excluded.

## Cell lines and cell culture

Fresh human PBMCs were cultured in RPMI(Gibco, cat#11875093) supplemented with 5% human serum (GemCell, cat#100-512), 12.5 mM HEPES (Gibco, cat#15630080), 2mM L-Glutamine (Gibco, cat#25030061), and 1X Pen/Strep (Gibco, cat#15140122). The jurkat report line pASP90 and pASP90-CD8 which are knocked out for endogenous TCR and express constitutive mCherry with GFP expression under the control of NFAT were a kind gift from Dr. Beatriz Careno[39]. Jurkat report cells, SUDHL10, and K562 cells were cultured in RPMI supplemented with 10% FBS (Gemini Bioproducts, cat#100-106), 2mM L-Glutamine, and 1X Pen/Strep. Patient-derived LCL cell lines were cultured in RPMI supplemented with 20% FBS, 2mM L-Glutamine, and 1X Pen/Strep. 293Ta cells were cultured in DMEM (Gibco, cat#11965092) supplemented with 10%FBS, 2mM L-Glutamine, and 1 mM sodium pyruvate (Gibco, cat#11360070). Tumor cell lines CAPAN1 (ATCC cat#HTB-79) and MDA-MB-231 (ATCC cat#CRM-HTB-26) were cultured in DMEM (Gibco cat#11965092) supplemented with 10% FBS, 2mM L-Glutamine, 1X Sodium pyruvate, and 1XPen/Strep.

## mKRAS-specific T cell tracking using IFNγ ELISPOT on fresh PBMC samples

The frequency of IFNγ secreting cells was determined using Human IFN-γ (HRP) precoated stripwells (Mabtech, cat#3420-4HST-2) following the Johns Hopkins Immune Monitoring Core SOPs. The tissue culture medium was composed of RPMI-1640 (Gibco, cat#11875-093) supplemented with L-glutamine (Mediatech, cat#25-005-Cl), antibiotics (Quality Biological, cat#120-095-721), 50 uM beta mercaptoethanol (Sigma, cat#) and 10% Fetal Calf Serum (Hyclone). The peptide pool control (CTL, cat# CTL-CEF-002) and KRAS peptides (JPT) were used at 2 ug/ml with 200,000 fresh PBMC responders per well and an 18-hour incubation time. Developed assays were read using

an iSpot Spectrum Analyzer (AID, Strassburg Germany) and reported as SFU (Spot Forming Units) per million cells.

## CyTOF of mKRAS peptide pulsed PBMCs

All antibodies used for CyTOF analysis are shown in Supplementary Table 10. Pre- and post-vaccine PBMCs were thawed at 37 °C and resuspended in complete media. PBMCs were seeded at $2 \times 10^6$ cell/well in 48 well plates and rested overnight. DMSO (Sigma Aldrich, cat#472301), control peptide, or mutant KRAS SLPs were added at 2ug/mL. CD3/CD28 Dynabeads (Gibco, cat#11161D) were used as a positive control. Cells with peptide were incubated for 48 h at 37 °C. eBioscience™ protein transport inhibitor cocktail (Thermofisher, cat#00-4980-93) was added to each well at 1X and cells were incubated for an additional 5 h. Cells were collected and transferred to a 96 well U bottom plate for staining. Cells were stained for viability with 20 µM cis-platinum (Standard BioTools, cat#201064) for 2.5 min at room temperature (RT), quenched with complete RPMI media, and washed twice in Maxpar Cell Staining Buffer (CSB, Standard BioTools, cat#201068). To ensure robust analysis with minimized batch effects, we employed 30-plex barcoding to stain all stimulation and timepoint conditions at the same time for each patient. A 7-choose-3 ($^{89}$Y, $^{110}$Cd, $^{111}$Cd, $^{112}$Cd, $^{113}$Cd, $^{114}$Cd, $^{116}$Cd) schematic was used based on CD45 to generate up to 35 unique barcodes. After staining for 25 min at RT, samples were washed three times in CSB and combined into batches via 40µm-strainer into 5-ml flow tubes. Batches were then incubated in Fc block (ThermoFisher Scientific, cat#14-9161-73) in CSB for 10 min at RT, stained with an antibody cocktail in CSB against chemokine receptors for 10 min at 37 °C, and then surface antibody cocktail in CSB for 30 min at RT. After two washes, batches were fixed/permeabilized using eBioscience™ Fix/Perm (ThermoFisher Scientific, cat#00-5123-43) for 30 min at RT. Cells were washed with eBioscience™ Perm Buffer twice and stained with intracellular staining cocktail for 30 min at RT. Fully stained cells were washed with CSB and fixed with freshly prepared Pierce™ paraformaldehyde 1.6% (ThermoFisher Scientific, cat#28906) in Maxpar PBS. On the day of acquisition, all cells were labeled with $^{103}$Rh (1:500, cat#201103, in Maxpar Fix & Perm Buffer, cat#201067, Standard BioTools) for 30 min at RT. Data was acquired on Helios™ at the Johns Hopkins Mass Cytometry Facility. Acquired data was preprocessed for normalization, randomization, and bead removal using CyTOF Software (v6.7, Standard BioTools). Gating for cell events, live cells, and debarcoding was performed using FlowJo (v10.9, BD). Processed FCS files were loaded into R (v4.0.2) for analysis. Expression levels were arcsine transformed with a cofactor of 5. Cells were clustered using FlowSOM algorithm and annotated based on canonical markers (e.g., CD3$^+$CD4$^+$CD8$^-$CD45RA$^+$CCR7$^+$CD45RO$^-$ characterizes naïve helper T cells)[50]. Cluster abundances were calculated as proportions of CD45$^+$ cells and were analyzed for significance relative to control peptide by one-way ANOVA followed by Dunnett's multiple comparisons test. Expression levels were compared using Wilcoxon test.

## Flow cytometry of mKRAS peptide pulsed PBMCs

PBMCs were thawed and rested overnight in complete media at $1 \times 10^6$ cell/well in a 96 well plate. The next day, DMSO control (unstimulated), the control peptide, or KRAS G12V, G12C, G12R, G12A, G12D, G13D 24-mer peptides were added to each well at a final concentration of 2ug/mL. PBMCs were incubated for 48 h. Six hours prior to collection, eBioscience protein transport inhibitor cocktail (Thermo Scientific, cat#00-4980-03) was added at a final dilution of 1X. Cells were collected and washed twice with PBS. Samples were incubated in Zombie NIR viability stain (Biolegend, cat#423105) for 15 min at room temperature in the dark. Samples were washed twice with FACS buffer containing 1x HBSS (Gibco, Serum 2%, Na Azide 0.1% and HEPES 0.1%). Cells were then stained with anti-CD3 (clone HIT3a Biolegend cat #300306), anti-CD4 (clone RPA-T4, Biolegend cat#300556), anti-CD8

(clone RPA-T8, Biolegend cat#301036), anti-CD45RO (clone UCHL1, cat#304218), anti-CD62L (clone DREG-56, Biolegend cat# 304823), anti-CD69 (clone FN50, Biolegend cat#310905), anti-CD137 (clone 4B4-1, Biolegend cat#309817) in FACS buffer. Samples were washed twice with FACS buffer and then permeabilized with BD Perm/Fix kit (BD Biosciences, cat# 554714) followed by staining with anti-IFNγ (clone 4S.B3, Biolegend cat#502512), anti-IL-2 (clone MQ1-17H12, Biolegend cat#500343), anti-TNFα (clone Mab11, Biolegend cat#502938), anti-T2A (clone 3H4, Novus cat#59627AF700). Samples were washed twice with 1X Perm/wash buffer and resuspended in FACS buffer. All samples were run on a Beckman Coulter Cytoflex cytometer. Data was analyzed in Flowjo version 10.8.1.

## Luminex cytokine measurements from supernatants of peptide restimulated PBMCs

The Bioplex 200 platform (Biorad,) was used to determine the concentration (pg/mL) of multiple target cytokines in plasma or cell culture supernatants. Luminex bead-based immunoassays (Millipore) were performed following the Johns Hopkins Immune Monitoring Core SOPs and concentrations were determined using 5 parameter log curve fits (using Bioplex Manager 6.0) with vendor-provided standards and quality controls. The HCYTA-60K panel was used to detect cytokines (IL-2, IL-4, IL-5, IL-10, IL-17A, IFNγ). Concentrations which were outside of the standard curves values were categorized as "out of range" (OOR). For each cytokine, OOR< values were replaced with the lower limit of the standard curve of the assay while OOR> were replaced with the upper limit of the standard curve.

## mKRAS-specific T cell in vitro expansion and TCR sequencing

PBMCs were thawed and rested overnight in complete media at $2 \times 10^6$ cell/well in a 48 well plate. The next day, DMSO control, the control peptide, or KRAS G12V, G12C, G12R, G12A, G12D, G13D 21-mer peptides were added to each well at a final concentration of 2ug/mL and 100IU/mL human IL-2 (Peprotech cat#200-02) and incubated at 37 °C for 7 days. Peptide expanded cells were collected, centrifuged at 1500 and genomic DNA was isolated using a DNeasy Blood and Tissue kit (Qiagen, cat#69504) according to the manufacturer's protocol. gDNA was quantified using a Nanodrop. TCRβ sequencing was performed by Adaptive Biotechnologies™.

## Tumor Bulk TCR Sequencing

Tumor DNA was extracted either from either FFPE slides or frozen core biopsies. For FFPE slides, scalpels were used for manual microdissection using slides stained with haematoxylin and eosin (H&E) as a guide. DNA was then extracted using the QIAmp DNA FFPE Tissue Kit (Qiagen #56404). TCR CDR3β sequencing was performed by Adaptive Biotechnologies. Frozen core biopsies were directly sent to Adaptive Biotechnologies for DNA extraction prior to sequencing.

## TCRβ repertoire analysis of expanded T cell populations

Shannon equitability for each expansion condition was calculated using the formula:

$$Shannon\ equitability = 1 - \frac{-\sum_{i=1}^{n} p_i log_e(p_i)}{log_e(n)}$$

where Pi is the frequency of an individual species and n is the total number of species. This value ranges between zero and one, with zero representing maximally even distributions and one representing maximum clonality[51,52]. TCRs expanded to any of the six described mKRAS peptides or control peptide were identified through comparison against DMSO control. Expanded clonotypes were identified using a one-sided fisher's exact test with false discovery rate correction using the Benjamini-Yekutieli procedure[53] in a manner similar to prior studies. Expanded clonotypes were identified using a one-sided fisher's

exact test with false discovery rate correction using the Benjamini-Yekutieli procedure[53] in a manner similar to prior studies[53–55]. Odds ratio (OR) is the absolute number of reads of a TCR clonotype found in an expansion condition over the absolute number of that respective TCRs in the control peptide condition[53]. Clonotypes were defined as the combination of TRBV and CDR3 amino acid sequence. Only expanded clonotypes with OR >= 5 over DMSO control and expansion frequency >= 0.1% were considered biologically meaningful and included for further analysis. Clones meeting expansion criteria for multiple mKRAS peptide stimulations were defined as cross-reactive. Expanded clones appearing in data for multiple subjects were considered public.

### Single cell TCR/RNA sequencing of unstimulated PBMCs

PBMCs were thawed on ice, washed, and resuspended in PBS containing 1% BSA, and cell counts and viability were made using Trypan Blue staining (ThermoFisher) in the hemocytometer. scRNA library preparations were performed using the 10x Genomics Chromium™ Single Cell system and Chromium™ Single Cell 5′ Library & Gel Bead Kit v2 (10x Genomics). TCRs were enriched using the TCR Amplification kit (10x Genomics), following the manufacturer's instructions. Initial cell input for each sample was 20,000 PBMCs to recover a total of 10,000 cells. Sequencing was performed on the NovaSeq platform (Illumina) using 10x Genomics recommended features and with a depth of 50,000 reads per cell. Samples were sequenced in two batches, with subject J1994_12 sequenced separately from remaining samples. Sequences were processed using the Cellranger 5.0.1 pipeline (10x Genomics) and mapped to the human reference genome (GRCh38).

### Single-cell RNA and TCR-seq analysis

Raw feature-barcode matrices were imported into Seurat version 4.3 for processing[56]. Samples from each of two sequencing batches were analyzed separately. Cells were clustered following QC using the Seurat *FindClusters* function using resolution 0.3. T cell clusters were differentiated from other PBMCs by expression patterns of positive and negative markers including CD3E, CD4, CD8B, CD14, CD19, FCGR3A, the presence of VDJ reads, and predicted cell type using Azimuth version 0.5.0 (PBMC reference, level 2 resolution)[56]. Following preliminary annotation, T cells were isolated and re-clustered. Cluster markers were identified using a Wilcoxon Rank Sum test as implemented through the Seurat *FindMarkers* function with default arguments including significance level = 0.01. Manual inspection of cluster markers and Azimuth predictions were used to make final cell type determinations. T cells with matching TRBV and CDR3 beta amino acid sequence to an anti-mKRAS TCR identified in expansions were considered anti-KRAS with peptide reactivity corresponding to expansion data. Gene expression analyses performed using native Seurat functions and custom analysis in R version 4.3.1.

### Generation of lentivirus and transduction of TCR expressing jurkat reporter cells

Complete paired TCRα and β sequences were compiled from the single cell dataset using StiTChR[57]. Third generation lentiviral vector plasmid (pLV) were designed to encode a TCRα-T2A-TCRβ expression cassette under control of the EF1α promoter. Plasmids were synthesized by VectorBuilder. To generate lentivirus, $3 \times 10^6$ 293Ta were seeded in a 10 cm dish with 10 mL complete media and incubated overnight 37 °C. 8ug pLV-TCR constructed with co-transfected into the cells with 6ug psPAX2 and 2ug pMD2.g using Lipofectamine 3000 (Thermo Fisher, cat#L3000001). Forty-eight hours later, supernatant was collected, centrifuged at 1500 rpm for 5 min, and filtered through a 0.45uM PES syringe filter (Celltreat, cat#229749). Virus was then concentrated using lenti-X concentrator (Takara

Bioscience, cat#631231) and resuspended in PBS. Concentrated virus was snap frozen in liquid nitrogen and stored at -80 °C.

To generate TCR expressing jurkat NFAT-GFP reporter cell lines, $1 \times 10^6$ cells were seeded in 0.5 mL complete media with 8ug/mL polybrene (Sigma-Aldrich, cat#H9268). 100–200ul of concentrated lentivirus was added to each well and cells were incubated overnight at 37 °C. The next day, 1 mL complete media was added to each well and cells were incubated for another 24 h at 37 °C.

Transduced cells were collected and washed twice with PBS. Cells were stained with Zombie NIR (1/1500 in PBS) and incubated for 15 min at room temperature in the dark. Cells were then washed twice in FACS buffer containing 1x PBS with 2% FBS. Anti-human TCRα/b-APC (clone IP26, Biolegend cat#306718) stain was added in FACS buffer to each well and incubated for 20 min at 4 °C. Cells were washed twice with FACS buffer and analyzed on a Beckman Coulter Cytoflex cytometer. Data was analyzed in Flowjo version 10.8.1. TCRα/β positive cells were then flow sorted on an BD Fushion sorter in sterile conditions.

### T cell engineering

Engineered primary human T cells expressing TCRs under control of an EF1-α promoter were generated via transduction with TCR-expressing lentivirus followed by CRISPR-Cas-mediated knock out of endogenous TCR[60]. Transgenic TCR sequences were codon optimized at the TRAC/TRBC locus to prevent CRISPR-Cas recognition and editing. T cells were isolated by negative selection using immunomagnetic cell separation (EasySep Human T Cell Isolation Kit #17951) from cryopreserved healthy donor peripheral blood mononuclear cells collected via leukapheresis. Purified CD3 + T cells were activated with Dynabeads Human T-Activator CD3/CD28 beads with 100 IU/mL recombinant human IL-2 (Peprotech #200-02) and 5 ng/mL recombinant human IL-7 (Peprotech #200-07) at 37 C, 5% $CO_2$. The next day, T cells were spinoculated with 100-200ul concentrated lentivirus at 920 g for 90 min at room temperature in the presence of 8 ug/mL polybrene (Sigma-Aldrich, cat#H9268). The next day, cells were collected and CD3/CD28 beads were removed with a magnet. Cpf1 (Cas12a) ribonucleoprotein (RNP) targeting *TRAC* (GAGTCTCTCAGC TGGTACAC) as well as *TRBC1* and *TRBC2* (GCCCTATCCTGGGTC-CACTC) were assembled by mixing the appropriate crRNA (IDT) with Alt-R A.s. Cas12a (Cpf1) Ultra nuclease (IDT) and Cpf1 Electroporation Enhancer (IDT) and incubating the mixture at room temperature for 15 min. To edit activated T cells, 20 μL of T cells were resuspended in P3 buffer at 5 ×107 cells/mL (Lonza) and added to the electroporation mixture. Electroporation was performed with a 4D-Nucleofector X Unit (Lonza) in 16-well cuvettes using pulse code EH115. After electroporation, T cells were recovered by immediately adding 80 μL of warm, cytokine-free T cell media to the cuvettes and incubation at 37 °C for 15 min. Then, T cells were diluted in T cell growth media containing 100 IU/mL recombinant human IL-2 and 5 ng/mL recombinant human IL-7. Cells were restimulated with Dynabeads Human T-Activator CD3/CD28 beads on day 8 and then expanded for 2 weeks replacing media with fresh cytokine media every 2 days for -two weeks prior to use in functional assays.

### Generation of patient matched LCL lines

$3 \times 10^6$ patient PBMCs from pre-vaccine time points were resuspended in 500ul human gammaherpesvirus 4 (HHV-4) (ATCC, cat# VR-1492), transferred to one well of a 24 well plate, and incubated for 2 h at 37 °C. After incubation, the plate was centrifuged at 1000 g for 10 min. 1.5 mL of LCL media was then added to each well. Cyclosporin A (Sigma-Aldrich, cat#C3662) was added to the cells at 0.1ug/mL. Cultures were incubated for 2-4 weeks at 37 °C to allow for LCL outgrowth. LCLs were phenotype by flow cytometry for expression of CD19, HLA class I, HLA class II, CD80, CD83, CD86, and CD137.

## Validation of mKRAS-specific T cell reactivity

Patient matched LCLs were collected and resuspended in Jurkat T cell media at a concentration of $1 \times 10^6$ cell/mL DMSO, control peptide, mutant KRAS or wildtype KRAS peptide (10uM-1pM) was added and cell were incubated at 37 C, 4 h. TCR expressing jurkat reporter cells were seeded at $1 \times 10^5$ cells/well in a 96 well plate. Peptide-pulsed LCLs were added to the jurkat cultures at a 1:1 APC:effector ratio. Cell stimulator cocktail was added to jurkat only cultures to determine the maximum GFP expression (%GFPmax). Co-cultures were incubated for 16–20 h at 37 °C. Co-cultures were then stained with Zombie NIR viability stain (1/1500 dilution, Biolegend, cat#423105) and analyzed by flow for mCherry and GFP expression as a marker for T cell recognition and activation. Co-cultured cells were analyzed for activation by GFP expression and normalized to specific activity of minimal (unpulsed LCL only) and maximal (PMA+Ionomycin) activation conditions.

## Tumor cell- T cell coculture assays

Upregulation of HLA class I and class II were first confirmed by flow cytometry staining of hIFNγ pre-treated tumor cells at 24, 48, or 72 h post-treatment. Based on this data, tumor cell lines were pretreated with 100IU/mL hIFNγ (peprotech #300-02) for 48 h. Cells were then washed with PBS and co-cultured with TCR transduced Jurkat cells or primary human T cells at an effector to target ratio of 2:1 for 24 h in the presence or absence of 2ug/mL supplemented mKRAS peptide. After incubation, jurkat co-cultures were analyzed by flow cytometry for upregulation of GFP as a marker of antigen recognition. For primary human T cells, expression of activation markers using anti-CD25 (clone BC96, Biolegend #302625) and anti-CD137 anti-CD137 (clone 4B4-1, Biolegend cat#309817) antibodies was assessed.

## Immunohistochemistry of primary tumor tissue

Immunostaining was performed at the Oncology Tissue Services Core of Johns Hopkins University School of Medicine. Immunolabeling for PD-L1, CD4, CD8, and HLA-DR,DP,DQ was performed on formalin-fixed, paraffin embedded sections on a Ventana Discovery Ultra autostainer (Roche Diagnostics). Briefly, following dewaxing and rehydration on board, epitope retrieval was performed using Ventana Ultra CC1 buffer (catalog# 6414575001, Roche Diagnostics) at 96 °C for 64 min. The following stains were performed:

For PD-L1 staining, anti- PD-L1 (1:100, catalog#13684, Cell Signaling Technology) was applied at 36 °C for 16 min. Primary antibodies were detected using an anti-rabbit HQ detection system (catalog# 7017936001 and 7017812001, Roche Diagnostics) followed by Chromomap DAB IHC detection kit (catalog # 5266645001, Roche Diagnostics), counterstaining with Mayer's hematoxylin, dehydration and mounting.

For CD4 and CD8 staining, primary antibody anti-CD8 (1:100 dilution; catalog# m7103, Dako) was applied at 36 °C for 60 min. CD8 primary antibodies were detected using an anti-mouse HQ detection system (catalog# 7017936001 and 7017782001, Roche Diagnostics) followed by Chromomap DAB IHC detection kit (catalog # 5266645001, Roche Diagnostics). Following CD8 detection, primary and secondary antibodies from the first round of staining were stripped on board using Ventana Ultra CC1 buffer at 93 °C for 8 min. Primary antibody, anti-CD4 (1:200 dilution; catalog# ab133616, Abcam) was applied at 36 °C for 20 min. CD4 primary antibodies were detected using an anti-rabbit HQ detection system (catalog# 7017936001 and 7017812001, Roche Diagnostics) followed by Discovery Teal Detection kit (catalog# 8254338001, Roche Diagnostics), counterstaining with Mayer's hematoxylin, dehydration and mounting.

For MHC class II staining, primary antibody, Anti-HLA DR + DP + DQ antibody (1:1000 dilution; catalog# ab7856 [CR3/43], Abcam) was applied at 36 °C for 60 min. Primary antibodies were detected using an anti-mouse HQ detection system (catalog# 7017936001 and 7017782001, Roche Diagnostics) followed by Chromomap DAB IHC detection kit (catalog # 5266645001, Roche Diagnostics), counterstaining with Mayer's hematoxylin, dehydration and mounting. Level of HLA-DR, DP, DQ expression on tumor cells in patient primary tumor resections was categorized as low, medium, or high expression by review by two individual expert pathologists.

## High similarity clonotypes in TCGA PDAC samples

To look for high similarity (HS-TCR) clonotypes, we used CDR3 amino acid sequences from PDAC tumor samples within the TCGA dataset reported in Thorsson et al.[58]. Using the expanded TCRβ repertoire from patients on this trial we detected TCR clonotypes with 2 or fewer amino acid differences using the stringdist R package (v 0.9.12). The KRAS mutations for TCGA PDAC samples were extracted from the TCGA Multi-Center Mutation Calling in Multiple Cancers project[59]. For every mKRAS positive PDAC samples from TCGA, we summarized the number of HS-TCR clonotypes.

## Reporting summary

Further information on research design is available in the Nature Portfolio Reporting Summary linked to this article.

## Data availability

All clinical, scRNAseq, scTCRseq, and bulk TCRseq data generated in this study have been deposited in the dbGap database under accession code phs003425 (https://www.ncbi.nlm.nih.gov/projects/gap/cgi-bin/study.cgi?study_id=phs003425.v1.p1). The CyTOF data generated in this study have been deposited at Zenodo at https://doi.org/10.5281/zenodo.15636498. Source data are provided with this paper.

## Code availability

All relevant code is available at (https://doi.org/10.5281/zenodo.11186766).

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

## Acknowledgements

This study was funded by National Cancer Institute grant F32CA271470-01 (ALH), National Cancer Institute grant U01CA253403 (EJF), National Cancer Institute Grant P01CA247886 (EMJ, ALH, and NZ), National Cancer Institute grant K08CA248624 (NZ), National Cancer Institute grant R50 CA243627 (LD), The Lustgarten Foundation (ALH, NZ, EMJ, EJF), Stand up to cancer (NZ, EJF, NZ, EMJ), ASCO Career Development Award (to NZ), The NIH Center Core grant P30 CA006973 (EMJ, EFDM, NZ), SPORE in Gastrointestinal Cancers P50CA062924 (EMJ, EFDM, NZ). We would like to thank our industry partners Bristol-Myers Squibb for supplying Ipi/Nivo and Oncovir for supplying the adjuvant poly-ICLC (Hiltonol®). We would also like to kindly thank Dr. Adham Bear, Dr. Bob Vonderheide and Dr. Beatriz Carreno for their guidance in experimental design and for sharing the pAS90 Jurkat TCR$_{KO}$ reporter line used in this study. We also thank the Investigational Drug Pharmacy at Johns Hopkins, in particular Hye Kim and Jacqueline Saunders, for their help in facilitating formulation of the vaccine. In addition, we would like to thank Vic Lemas (Johns Hopkins) and Marcia Meseck (Mount Sinai) for their advice on vaccine formulation. We would like to acknowledge M. Popoli, J. Ptak, N. Silliman, L. Dobbyn, and B. Vogelstein for performing and/or interpreting ctDNA assays.

## Author contributions

Conceptualization: N.Z., E.M.J., N.S.A., E.J.F., W.J.H., A.L.H., A.A.G. Clinical: S.D.H., T.H., M.B., K.B., M.B., M.Y., D.L., N.S.A., N.Z. Regulatory: J.D., J.M.N., A.M.T. Statistical analysis: H.W. (Hao Wang), J.L. Experimental Design: A.L.H., N.Z., E.M.J., E.J.F., A.A.G., Y.W., C.T., L.T.K., M.F.K., B.M., J.M., W.J.H. E.J.F., E.M.J., N.Z. Data collection: A.L.H., L.A., A.H., G.L., Y.W., B.B., C.T., B.M., L.T.K., Z.Z., M.F.K., B.J.M. Data analysis: A.L.H., A.A.G., S.D.H., H.W. (Henry Wang), L.D., E.D.M., Y.W., D.H.S., J.W.L., R.A.A., M.F.K., B.J.M., E.C.M., W.J.H., E.M.J., N.Z. Software: A.A.G., E.F.D.M., L.D., J.M., L.T.K., W.J.H., E.J.F. Supervision: N.Z., E.M.J., N.S.A., E.J.F., W.J.H., L.T.K., D.L. Manuscript preparation: A.L.H., S.D.H., A.A.G., L.D., E.M.J., N.Z. Manuscript review: A.L.H., S.D.H., A.A.G., L.D., Y.W., L.A., G.L., J.W.L., D.H.S., J.D., J.M.N., J.L., S.M., H.W., E.J.M., W.J.H., N.S.A., E.M.J., N.Z.

## Competing interests

E.M.J. reports other support from Abmeta, other support from Adventris, personal fees from Achilles, personal fees from DragonFly, personal fees from Parker Institute and CPRIT, personal fees from Surge and HDTbio, grants from Lustgarten, grants from Genentech, personal fees from Mestag, personal fees from Medical Home Group, grants from BMS, and grants from Break Through Cancer outside the submitted work. Dr. Jaffee is the Dana and Albert "Cubby" Broccoli Professor of Oncology. ALH is a paid consultant of Adventris Pharmaceuticals Inc. SDH reports consulting for Sift Biosciences and HMP Omnimedia; research support from Summit Therapeutics, Shionogi, and Bristol Myers Squibb; and travel/honoraria from DAVA Oncology. NZ receives research support from Bristol Myers Squibb, is a consultant for Genentech, and receives other support from Adventris Pharmaceuticals. NSA is a paid consultant for Mirati and QED. Dr. Azad receives institutional funding from Agios, Inc., Array, Atlas, Bayer HealthCare, BMS, Celgene, Debio, Eli Lilly and Company, EMD Serono, Incyte Corporation, Intensity, Merck & Co., Inc. and Taiho Pharmaceuticals Co., Ltd. Dr. Azad participates on advisory boards for Incyte, QED, and Glaxo Smith Kline. EJF is on the scientific advisory board of Resistance Bio, received consulting fees from Mestag Therapeutics and Merck, and receives research funding from Abbvie, Inc. 'WJH reports patent royalties from Rodeo/Amgen, received grants from Sanofi and NeoTX (to Johns Hopkins), and speaking/travel honoraria from Exelixis and Standard BioTools. N.Z., E.M.J., A.L.H., M.Y., A.A.G., and E.F.J. have current filed patents related to the KRAS peptides and/or TCRs. The remaining authors declare no competing interests.

## Additional information

[1]Johns Hopkins Convergence Institute, Johns Hopkins University School of Medicine, Baltimore, MD, USA. [2]Johns Hopkins Bloomberg Kimmel Institute for Immunotherapy, Johns Hopkins University School of Medicine, Baltimore, MD, USA. [3]Department of Oncology, Sidney Kimmel Comprehensive Cancer Center, Johns Hopkins University, Baltimore, MD, USA. [4]Department of Gastrointestinal Medical Oncology, The University of Texas MD Anderson Cancer Center, Houston, TX, USA. [5]Department of Genetic Medicine, Johns Hopkins University School of Medicine, Baltimore, MD, USA. [6]Department of Pathology, Johns Hopkins University School of Medicine, Baltimore, MD, USA. [7]Greenebaum Comprehensive Cancer Center, University of Maryland School of Medicine,

Baltimore, MD, USA. [8]Ludwig Center, Sidney Kimmel Comprehensive Cancer Center, Johns Hopkins University School of Medicine, Baltimore, MD, USA. [9]Lustgarten Pancreatic Cancer Research Laboratory, Sidney Kimmel Comprehensive Cancer Center, Johns Hopkins University School of Medicine, Baltimore, MD, USA. [10]Howard Hughes Medical Institute, Chevy Chase, MD, USA. [11]Division of Rheumatology, Department of Medicine, Johns Hopkins University School of Medicine, Baltimore, MD, USA. [12]These authors contributed equally: Amanda L. Huff, S. Daniel Haldar. ✉e-mail: ejaffee@jhmi.edu; nzaidi1@jhmi.edu

