## [Transparent Peer Review file · Nature Communications]

A pooled mutant KRAS peptide vaccine and immune checkpoint blockade activates diverse T cell responses in resected PDAC

Corresponding Author: Dr Neeha Zaidi

Version 0:

Reviewer comments:

Reviewer #1

(Remarks to the Author)

The authors have addressed my relatively minor queries.

Supplemental Fig 7 is particularly helpful new information and even with an N of 1 (patient 5) provides confidence that the majority of KRAS targeted TCRs are generated de novo post vaccination.

Reviewer #3

(Remarks to the Author)

Comments

In Supplemental figure 3g, the definitions of what are considered “low”, “moderate”, and “high” levels of HLA-DR,DP,DQ expression are not provided. Please clarify or show example staining/quantification.

In Figure 2d, it may be helpful to also include asterisks above the G12R responders for clarity.

In Figure 6g, in MDA-MB-231 third flow plot, please make clear you are adding KRAS G13D as you noted in the CAPAN1 + KRAS G12V.

In lines 401-402, referring to figure 6h: “After co-culture with KRAS G12V+ CAPAN1 cells pre-treated with IFN γ , we observed a modest increase in CD25 and CD137 expression relative to untreated CAPAN1 cells.” This claim appears to be overstated and should likely be removed if it is indeed referring to the double positive population rising from 0.18 in untreated to 0.38 in the IFN γ treated.

In Supplemental Figure 9f, please label the frequency of cells present in the contour plots for clarity.

Minor formatting/editing comments

There appears to be a formatting error in Supplemental figure 4d.

Line 337 should read “Supplemental Data 5”

Line 348 should read “Supplemental Figure 8i”

Line 369 should read “Figure 6e”

We thank the reviewers for their thoughtful and constructive comments and have provided a point-by-point response in blue. Revision in line call outs here refer to the mark up version of our revised manuscript.

Reviewer #1

(Remarks to the Author)

In this important study, Huff and colleagues have conducted a trial of pooled synthetic long peptide (SLP_ vaccines against mKRAS in patients who underwent surgery for localized pancreatic cancer. Vaccines were administered agnostic to patient HLA and were initially combined with immune checkpoints (aPD1 + aCTLA4) for the first four doses. The majority of patients (10/12) generated significant tumor specific T cell responses in the periphery, which were dominated by a Th1 CD4 T cell response. The authors identified distinctions in the amplitude of immunogenic T cell responses based on mKRAS allelic variant, with G12V and G12R generating higher immunogenic responses than G12D or G13D (agnostic to the dominant allelic variant in the resected primary tumor). Importantly, patients generated both mono-reactive and cross-reactive T cell responses to the SLP vaccine. One critical and valuable resource developed by this manuscript is the identification of a large panel of mKRAS specific TCR sequences, both private (patient-specific) and public, which can be deployed in future for generating engineered T cells for adoptive therapy. Remarkably, four of 12 patients had recurrence free survival at 52 weeks of follow up at which point they entered an extended vaccine only phase of therapy. AT >100 weeks, all four patients had demonstrate mKRAS specific T cells in the circulation, with Cd4+ T cells dominating in the response.

This study establishes the safety and feasibility, as well as early efficacy of a pooled mKRAS targeted SLP vaccine in resected pancreatic cancer. There is extensive phenotyping of T cell responses post vaccination (including in the long term responders) and cataloging of an extensive cache of mKRAS specific TCRs (a subset with functional validation).

I only have some minor queries, mostly aligning this study in the context of previously published trials in this space using mRNA and amphiphile vaccines.

Response: The authors are grateful for the reviewer's thoughtful and thorough appraisal of our manuscript and for the insightful comments.

A: It is unclear if neoadjuvant therapy versus upfront surgery had an effect on DFS. In Fig 1B, 3 of the 4 long term responders (patients 1, 5 and 14) received neoadjuvant therapy. As a corollary, please confirm that patient 2 does not harbor microsatellite instability or a germline alteration like BRCA2 that might render the tumor more "immunogenic".

Response: We have performed an analysis on the impact of neoadjuvant chemotherapy and DFS and find no difference in outcomes in those who do versus those who do not receive neoadjuvant chemotherapy. This is now added as a new Supplemental Figure 2i below. We also note that the benefit of neoadjuvant chemotherapy on long-term

outcomes in resected disease is unclear with mixed data reported (PREOPANC, ESPAC-5F trials).

New Supplemental Figure 2i:

This is also referenced in the text on lines 169-172 as follows:

“There was no correlation between DFS and lymph node status, primary tumor size, baseline CA19-9 levels, ctDNA positivity at baseline, neoadjuvant chemotherapy, splenectomy, or absolute lymphocyte counts at baseline (Supplemental Figure 2e-k).”

Additionally, we can confirm that patient 2 does not harbor microsatellite instability or a pathogenic germline alteration, such as BRCA2. All patients had previously undergone next-generation sequencing of their tumor tissue, and all of the tumors were TMB low. We have added this information on TMB and MSI status into our new Supplemental Table 1 for clarity:

New Supplemental Table 1: Patient demographics and clinical characteristics

Characteristic	All Patients (n = 12)	Recurrence (n = 7)	No Recurrence (n = 5)	P Value*
Age (years)				
Mean (SD)	64.2 (10.5)	63.3 (8.9)	65.4 (13.5)	0.67
Median (range)	68.0 (42.0-76.0)	59.0 (53.0-76.0)	69.0 (42.0-75.0)	
Sex				0.15
Male	10 (83.3%)	7 (100%)	3 (60%)	
Female	2 (16.7%)	0 (0%)	2 (40%)	
Race				0.47
White	10 (83.3%)	5 (71.4%)	5 (100%)	
Asian	2 (16.7%)	2 (28.6%)	0	
Tumor Location				

Head	7 (58.3%)	3 (42.9%)	4 (80%)	0.12
Body	4 (33.3%)	4 (57.1%)	0	
Tail	1 (8.3%)	0	1 (20%)	
Tumor Size (cm)				
Mean (SD)	2.78 (0.88)	3.1 (0.9)	2.4 (0.8)	0.25
Median (range)	2.50 (1.50-4.50)	3.0 (2.2-4.5)	2.4 (1.5-3.5)	
Tumor Differentiation				
Well-to-moderately differentiated	1 (8.3%)	0	1 (20%)	0.046
Moderately differentiated	7 (58.3%)	6 (85.7%)	1 (20%)	
Moderate-to-poorly differentiated	2 (16.7%)	1 (14.3%)	1 (20%)	
Poorly differentiated	2 (16.7%)	0	2 (40%)	
Pathologic Stage				
IA	1 (8.3%)	0	1 (20%)	0.85
IB	2 (16.7%)	1 (14.3%)	1 (20%)	
IIB	6 (50.0%)	4 (57.1%)	2 (40%)	
III	3 (25.0%)	2 (28.6%)	1 (20%)	
Lymph Node Status				
N0	3 (25.0%)	1 (14.3%)	2 (40%)	0.77
N1	6 (50.0%)	4 (57.1%)	2 (40%)	
N2	3 (25.0%)	2 (28.6%)	1 (20%)	
Resection Margins				
R0	11 (91.7%)	7 (100%)	4 (80%)	0.42
R1	1 (8.3%)	0	1 (20%)	
Neoadjuvant Chemotherapy				
(m)FOLFIRINOX	6 (50%)	3 (42.9%)	3 (60%)	>0.99
None	6 (50%)	4 (57.1%)	2 (40%)	
Adjuvant Chemotherapy				
(m)FOLFIRINOX	8 (66.7%)	5 (71.4%)	3 (60%)	0.82
Gemcitabine/nab-paclitaxel	1 (8.3%)	1 (14.3%)	0	
Gemcitabine/cisplatin/nab-paclitaxel	1 (8.3%)	0	1 (20%)	
None	2 (16.7%)	1 (14.3%)	1 (20%)	
Perioperative Radiation				
Yes	3 (25%)	1 (14.3%)	2 (40%)	0.52
No	9 (75%)	6 (85.7%)	3 (60%)	
TMB				
Low (<10 mutations per Mbp)	12 (100%)	7 (100%)	5 (100%)	>0.99
High (>10 mutations per Mbp)	0 (0%)	0 (0%)	0 (0%)	
Microsatellite instability (MSI) status				
MSS (microsatellite stable)	11 (100%)	6 (50%)	5 (100%)	>0.99
MSI- high	0 (0%)	0 (0%)	0 (0%)	
MSI- low	0 (0%)	0 (0%)	0 (0%)	

Unknown (not evaluable)	1 (8.3%)	1 (8.3%)	0 (0%)	
Baseline CA-19-9 (U/mL)				
Mean (SD)	24.8 (21.2)	26.3 (24.8)	17.7 (17.0)	
Median (range)	26.1 (0-69.9)	26.1 (0-69.9)	12.7 (0-42.1)	0.75
KRAS Mutation				
G12D	4 (33.3%)	4 (57.1%)	0	0.086
G12R	2 (16.7%)	1 (14.3%)	1 (20%)	
G12V	6 (50%)	2 (28.6%)	4 (80%)	
Time From Completion of Adjuvant Chemotherapy to First mKRAS-VAX (months)				
Mean (SD)	3.9 (1.7)	3.5 (1.5)	4.4 (1.9)	
Median (range)	4.2 (1.5-6.0)	3.4 (1.5-5.4)	5.0 (1.6-6.0)	0.25

B: The 82% response rate of KRAS targeted T cell responses observed in this study aligns almost perfectly with the responses observed in the amphiphile peptide vaccine trial (Elicio, Pant et al Nat Med 2024), but both are higher than mRNA vaccines (~50%, Rojas Nature 2023). While the small numbers make these comparisons fraught, do the authors believe that peptide vaccines might be more immunogenic than mRNA vaccines. In particular, the dominance of a CD4+ T cell response stands out here versus CD8+ T cell responses with mRNA vaccines.

Response: We appreciate this comment and agree that there is a need to compare the mRNA and peptide vaccine platforms head-to-head to assess differences in immunogenicity, particularly as it relates to mutant KRAS neoantigens. We do note, however, in Rojas *et al.*, that the subset of patients who were vaccinated with mutant KRAS antigens did not generate a T cell response to mKRAS. Additionally, we agree that vaccination with SLPs have consistently generated CD4 T cell responses (Ott *et al.*, Nature 2017, Ott *et al.*, Cell 2020, Pant *et al.*, Nature Medicine 2024). We have added these points to paragraph 4 of the discussion starting on line 464 as follows:

“Cancer vaccines must not only generate a diverse activated T cell response for tumor killing but also elicit memory T cells should recurrence occur. We show, through single cell proteomics and transcriptomics, that mKRAS-VAX activates mKRAS-specific Th1 CD4 central and effector memory T cells that express functional cytokines IL2, IFN γ and TNF α as well as cytotoxic effector memory CD8 T cells which express the cytotoxic effector molecules GzmB. The Th1 CD4 responses observed are consistent with other peptide-based vaccine approaches and have previously been associated with improved clinical outcomes³¹. Notably, and consistent with our approach, the amphiphile-peptide mKRAS vaccine also elicited predominantly Th1 CD4 T cells, while in contrast no mKRAS-specific T cells were detected by mRNA-based vaccine. Additionally, the mRNA-based approach mounted predominantly CD8 T cells to the targeted personalized neoantigens, highlighting the need for more study of the differences of T cell populations activated by each platform. Lower levels of the Th2-associated cytokine IL-5, but not IL-4, were also

detected after mKRAS peptide stimulation in this study. While IL-5 function is still not entirely understood in the context of immunotherapy, recently, IL-5 secreting CD4 T cells have been associated with efficacy of ICB by promoting eosinophil expansion, tumor infiltration, and thus intra-tumoral CD8 T cell activation in breast cancer³². In our study, we found that the activated T cell populations persisted and tracked with a durability marker CCR5 in the four patients who remained enrolled during the extended vaccine-only, no-ICI phase. The CD4 Tem population was the most durable, likely owing to the trafficking of CD4 Tem to the secondary lymphoid tissues over time due to CCR7 expression. Furthermore, granzyme B was detected in responding mKRAS-specific CD4 T cells, questioning the potential for a role of CD4 T cells in direct tumor killing. In the small subset of TCRs validated in our study, T cells recognized antigen when peptide was supplied exogenously or when KRAS-mutation specific tumor cell lysate was pulsed onto APCs, suggesting their function in controlling tumor growth may be more related to indirect mechanisms of tumor control as previously reported³³⁻³⁷. Notably, Alspach *et al* described the critical role of neoantigen-specific CD4 T cell-mediated control of tumor growth in tumors that lacked expression of MHCII, underscoring the potential importance of these T cell responses³³⁻³⁷. We also found that four patients who remained recurrence free displayed moderate levels of HLA-DR, DP and DQ in their primary tumor tissues, again suggesting that CD4 T cells may directly recognize patient tumors. However, additional testing of mKRAS-specific CD4 T cells in our dataset and their ability to lyse tumor directly needs to be explored. Nonetheless, these data appear to be consistent with high levels of HLA class II associated with greater activation of T regulatory cells and a more immunosuppressive TME³⁸.”

C: One plausible hypothesis that was proposed in the Rojas et al study was the possibility of splenectomy dampening T cell responses to the vaccine. Did the authors observe any effect of splenectomy on intensity or duration of responses.

Response: We thank the reviewer for this comment. We did not find any significant association between the magnitude of mKRAS-specific T cell responses and splenectomy status in this cohort. These data have been included in the manuscript as new Supplemental Figure 2j as follows:

This is also referenced in the text on lines 169-172 as follows:

“There was no correlation between DFS and lymph node status, primary tumor size, baseline CA19-9 levels, ctDNA positivity at baseline, neoadjuvant chemotherapy, splenectomy, or absolute lymphocyte counts at baseline (Supplemental Figure 2e-k).”

D: The differences in mKRAS allele specific T cell responses for G12V and G12R versus G12D are truly remarkable. Could the authors confirm that patients with a "tumor discordant" major T cell response (for example patient 18 - a G12D tumor with significant G12R and G12V responses) did not harbor these mutant clones as secondary mKRAS alterations within the resected tumor? And that this is simply an example of epitope spreading?

Response: We can confirm that we did not see any other secondary mKRAS alterations within the resected tumor. As part of our ctDNA analysis, we performed additional whole exome sequencing on 9 patient tumor samples for those with enough available tissue. This analysis did include patient 18's primary tumor, which only expressed a detectable KRAS G12D mutation as shown in Revision Table 1. While we cannot exclude epitope spreading, we do vaccinate with all six mKRAS antigens and thus would expect generation of T cell responses across the majority of vaccinated neoepitopes. As this data was consistent with our CLIA-certified WES performed in the clinic (reported in Supplemental Table 1 and Supplemental Table 6 for each patient), we provide our additional confirmation analysis as Revision Table 1, below:

Revision Table 1: Summary of WES detection of KRAS mutations in primary tumor

Patient ID	KRAS mutations detected	%MAF, WBC	%MAF, tumor
10	p.G12D	0%	32%
9	p.G12V	0%	19%
1	p.G12R	0%	9%
2	p.G12V	0%	25%
14	p.G12V	0%	3%
3	p.G12D	0%	20%
6	p.G12V	0%	8%
7	p.G12V	0%	14%
18	p.G12D	0%	13%

E: As a corollary to the point above, one missing link here is the demonstration of whether these mKRAS reactive clonotypes were present in the tumor tissue (not PBMCs) at baseline. The entire study is based on baseline and post vaccine PBMCs, but excluding

the possibility of existence of these clonotypes with some form of bulk TCR assay on tumor tissue would be helpful to validate the vaccine is not expanding pre-existing clones.

Response: To assess if these mKRAS reactive clonotypes were present in tumor tissue at baseline, we performed bulk TCR β sequencing on FFPE serial sections of the primary tumors from 3 out of the 4 patients for whom sufficient tumor sample was available. We also performed TCR β sequencing on a recurrence lesion from one of the patients that was available. These data are presented in Supplemental Figure 7D-F. When mKRAS-specific TCRs were mapped onto the tumor TCR repertoires, few mKRAS-specific TCRs were found in the primary tumors of these patients relative to a recurrence biopsy sample available for patient 5. For example, only 1 unique mKRAS clonotype was observed in the baseline tumor sample for patient 5, while 80 unique clonotypes were observed in the post-vaccination, recurrence lesion. To clarify this, we have added a figure that shows the percent of mKRAS TCRs found in the tumor tissue at baseline for patients 2, 5, and 6 as well as a post-vaccine timepoint for patient 5. We observe the majority of mKRAS TCRs identified are found only in the post-vaccine sample for patient 5. These data, in combination with our PBMC expansion data, offer support for the conclusion that the majority of the mKRAS clonotypes detected were *de novo* and not found in the blood or primary tumor samples of these patients prior to vaccination.

New Supplemental Figure 7D-F:

This is also referenced in the revised manuscript on lines 275-282 as follows:

“Consistent with this, there was little overlap between the mKRAS-specific TCRs identified by expansion and bulk TCR β sequencing of the resected primary tumors, despite the high numbers of productive TCRs sequenced for all tissue specimens (**Supplemental Figure 7d,e**).

Conversely, a single metastatic lung recurrence biopsied from patient J1994_5 had 80 unique mKRAS-specific TCR sequences overlapping within the tissue representing almost 20% of the mKRAS-repertoire identified for this patient (**Supplemental Figure 7e**). Of the mKRAS-specific TCRs in the tumor, 55% were specific for the patient’s tumor mutation (G12V) (**Supplemental Figure 7f**).”

Reviewer #2

(Remarks to the Author)

In this study, Huff et al conducted a phase 1 trial in 12 patients with resected pancreatic ductal adenocarcinoma (PDAC), administering a vaccine of a pool of six KRAS mutant long peptides (mKRAS) in combination with immune checkpoint inhibitors. They did not find significant vaccine-related adverse events and found that majority of patients generated a significant increase in average T cell response to the 6 mKRAS antigens included in the vaccine, and tumor-specific mKRAS response. Authors were able to identify the generation of CD4 and CD8 T cells with distinct phenotypes and both mono- and cross-reactive T cell responses against mKRAS antigens.

The study is well conducted and elaborate, however, most of the reported findings lack novelty and highly predictable based on many vaccine trials conducted over the past 20 years. This include lack of novelty in the conduct of a vaccine trial with pool of peptides, the ability of the patients to generate CD8 or CD4 immune responses against each of the antigens separately, cloning of TCRs against specific ras antigenic epitopes have been reported and also used in T cell therapy and finally, the clinical findings are highly predictable. Furthermore, there are some concerns:

Response: We appreciate the reviewer comments and agree that prior KRAS mutation targeted vaccines had been tested years ago before immune checkpoint inhibitors were available, and many were against HLA-restricted peptides. While some of these did show T cell responses in the blood, the technologies to conduct TCR sequencing and multiplex analyses were not available. Thus, the use of these technologies to obtain granular details of immune response separates our study from the prior ones. We also build upon previous studies by:

1. In this study, we are the first to directly compare the immunogenicity profile of the 6 most common mutant KRAS antigens in a population of non-HLA restricted patients. Thus, we were able to directly identify the weaker immunogenicity of KRAS G12D relative to other KRAS antigens. These findings have an important biological context for future vaccine designs targeting mKRAS antigens, as well as, broadly to the field studying the immune profile of these antigens.
2. Our results establish the largest repertoire of mutant KRAS-specific T cells identified across patients and against the 6 most KRAS mutations to date. This dataset has significant implications for further understanding of mutant KRAS T cell biology across the six most common KRAS driver neoantigens, and more generally, for the field of adoptive therapy targeting these antigens.
3. Few, if any studies have shown the prevalence of common T cell receptors recognizing the same antigen across patients treated with a neoantigen vaccine, or indeed, T cell receptor sequences that recognize more than one mutation within the same original antigen. These data support the feasibility of a pan-KRAS “off-the-shelf” adoptive T cell approach for future testing in patients with KRAS-mutation-driven tumors (about 20% of all cancers).

4. Our study is the first to test a KRAS peptide vaccine plus anti-PD-1 and anti-CTLA-4 antibodies in an adjuvant setting for pancreatic adenocarcinoma (PDAC). Based on our prior preclinical studies, this combination is critical for inducing the highest quality of neoantigen-specific T cells.
5. While this study represents the first testing of this off-the-shelf vaccine including all 6 mKRAS antigens for immunogenicity in an adjuvant setting, the data provide the basis for its testing as a preventative vaccine for patients who are at high risk of PDAC, currently underway [NCT04117087] – the latter representing the first testing of any cancer vaccine in a PDAC high-risk population. For this, we were required to establish safety and immunogenicity in the therapeutic setting with this exact vaccine platform before vaccinating otherwise healthy individuals. As high-risk patients do not yet have cancer, single mutant KRAS antigen vaccination would not be an effective vaccine approach – thus our rationale for a pooled peptide approach that provides the broadest coverage against mKRAS antigens in a vaccine yet.

Major concerns:

The study is unable to draw any definitive conclusions about clinical efficacy (e.g., DFS differences between mutation subtypes). The safety of peptide vaccines has been exhaustively reported, and it was also published in a comprehensive review (Rahma et al. Clin Cancer Res. 2014). Furthermore, although the authors acknowledge these causal inferences regarding improved disease-free survival (DFS) remain speculative in a a) single arm, b) small sample size of 12 patients, c) in combination therapy and d) with a limited follow-up duration. It's impossible to disentangle the Contribution of the vaccine versus natural immune recovery or checkpoint blockade effects.

Response: We agree with the reviewer that this is a small, phase I trial to establish safety and immune responses of our pooled peptide platform as a prelude to testing in a prevention setting in high-risk individuals. While the association between T cell responses and DFS are correlative, the signal we observed is consistent with recent studies in resected PDAC targeting either mKRAS neoantigens (Pant *et al.* Nature Medicine, 2024) or personalized neoantigens (Rojas *et al.* Nature, 2023). Furthermore, it is well known that immune checkpoint blockade alone does not have efficacy in the vast majority of PDACs either as single or double agents. To address this, we have expanded on our discussion of the limitations of this study starting on line 428 as follows:

“Patients were allowed to enroll within 6 months of completing adjuvant chemotherapy and therefore joined the study at different timepoints within this window. Our exploratory analysis suggests that patients in the top three quartiles of average tumor-specific T cell responses (by ELISPOT) experienced greater DFS, which we have defined from the time of first vaccination. Notably, we did not observe a correlation of DFS with patient characteristics, tumor size, lymph node status at resection, CA-19-9 levels, ctDNA positivity, neoadjuvant chemotherapy,

splenectomy, or lymphocyte count at baseline. Although perhaps informative for a potential magnitude of response that is required for clinical benefit, these correlations should be interpreted with caution given that the trial was not powered for clinical efficacy but appears to be consistent with recent neoantigen vaccine studies in PDAC^{19,22}. Our study was also limited in analysis of post-vaccine tumor samples which were unavailable in this cohort. Future studies which correlate mKRAS-specific T cell infiltration into tumors with clinical outcomes may also be more informative.”

We have also provided clinical information from the cohort that all patients were TMB low and microsatellite stable supportive that ICB alone would likely be ineffective in this setting. This data is now included in new Supplemental Table 1 as follows:

Characteristic	All Patients (n = 12)	Recurrence (n = 7)	No Recurrence (n = 5)	P Value*
Age (years)				
Mean (SD)	64.2 (10.5)	63.3 (8.9)	65.4 (13.5)	0.67
Median (range)	68.0 (42.0-76.0)	59.0 (53.0-76.0)	69.0 (42.0-75.0)	
Sex				0.15
Male	10 (83.3%)	7 (100%)	3 (60%)	
Female	2 (16.7%)	0 (0%)	2 (40%)	
Race				0.47
White	10 (83.3%)	5 (71.4%)	5 (100%)	
Asian	2 (16.7%)	2 (28.6%)	0	
Tumor Location				0.12
Head	7 (58.3%)	3 (42.9%)	4 (80%)	
Body	4 (33.3%)	4 (57.1%)	0	
Tail	1 (8.3%)	0	1 (20%)	
Tumor Size (cm)				0.25
Mean (SD)	2.78 (0.88)	3.1 (0.9)	2.4 (0.8)	
Median (range)	2.50 (1.50-4.50)	3.0 (2.2-4.5)	2.4 (1.5-3.5)	
Tumor Differentiation				0.046
Well-to-moderately differentiated	1 (8.3%)	0	1 (20%)	
Moderately differentiated	7 (58.3%)	6 (85.7%)	1 (20%)	
Moderate-to-poorly differentiated	2 (16.7%)	1 (14.3%)	1 (20%)	
Poorly differentiated	2 (16.7%)	0	2 (40%)	
Pathologic Stage				0.85
IA	1 (8.3%)	0	1 (20%)	
IB	2 (16.7%)	1 (14.3%)	1 (20%)	
IIB	6 (50.0%)	4 (57.1%)	2 (40%)	
III	3 (25.0%)	2 (28.6%)	1 (20%)	

Lymph Node Status				
N0	3 (25.0%)	1 (14.3%)	2 (40%)	0.77
N1	6 (50.0%)	4 (57.1%)	2 (40%)	
N2	3 (25.0%)	2 (28.6%)	1 (20%)	
Resection Margins				
R0	11 (91.7%)	7 (100%)	4 (80%)	0.42
R1	1 (8.3%)	0	1 (20%)	
Neoadjuvant Chemotherapy				
(m)FOLFIRINOX	6 (50%)	3 (42.9%)	3 (60%)	>0.99
None	6 (50%)	4 (57.1%)	2 (40%)	
Adjuvant Chemotherapy				
(m)FOLFIRINOX	8 (66.7%)	5 (71.4%)	3 (60%)	0.82
Gemcitabine/nab-paclitaxel	1 (8.3%)	1 (14.3%)	0	
Gemcitabine/cisplatin/nab-paclitaxel	1 (8.3%)	0	1 (20%)	
None	2 (16.7%)	1 (14.3%)	1 (20%)	
Perioperative Radiation				
Yes	3 (25%)	1 (14.3%)	2 (40%)	0.52
No	9 (75%)	6 (85.7%)	3 (60%)	
TMB				
Low (<10 mutations per Mbp)	12 (100%)	7 (100%)	5 (100%)	>0.99
High (>10 mutations per Mbp)	0 (0%)	0 (0%)	0 (0%)	
Microsatellite instability (MSI) status				
MSS (microsatellite stable)	11 (100%)	6 (50%)	5 (100%)	>0.99
MSI- high	0 (0%)	0 (0%)	0 (0%)	
MSI- low	0 (0%)	0 (0%)	0 (0%)	
Unknown (not evaluable)	1 (8.3%)	1 (8.3%)	0 (0%)	
Baseline CA-19-9 (U/mL)				
Mean (SD)	24.8 (21.2)	26.3 (24.8)	17.7 (17.0)	0.75
Median (range)	26.1 (0-69.9)	26.1 (0-69.9)	12.7 (0-42.1)	
KRAS Mutation				
G12D	4 (33.3%)	4 (57.1%)	0	0.086
G12R	2 (16.7%)	1 (14.3%)	1 (20%)	
G12V	6 (50%)	2 (28.6%)	4 (80%)	
Time From Completion of Adjuvant Chemotherapy to First mKRAS-VAX (months)				
Mean (SD)	3.9 (1.7)	3.5 (1.5)	4.4 (1.9)	0.25
Median (range)	4.2 (1.5-6.0)	3.4 (1.5-5.4)	5.0 (1.6-6.0)	

2. Authors fail not find any correlation between DFS and the densities or proportions of CD4, CD8 T cells, macrophages, or PD-L1 expression within the primary tumor. However,

in a recent study in pancreatic and colorectal cancer, an association between relapse-free survival (RFS) and baseline absolute lymphocyte count was reported (Nature Med, 2024, doi:10.1038/s41591-023-02760-3). Authors should discuss the potential reasons for these differences between the two trials, especially the type of vaccine/formulation, patient selection criteria and how these may affect the prediction, treatment outcome or the application of a particular vaccine in patients with mKRAS PDAC.

Response: We appreciate this comment, and, in response, we have performed an analysis to assess a possible association between DFS, and baseline absolute lymphocyte count in the periphery, as reported in the recent manuscript (Pant *et al*, Nature Medicine 2024). In this manuscript, the authors suggest this correlation meant that “complete hematologic recovery from prior cytotoxic therapy may be required for optimal vaccine response”. In contrast, we had performed correlations by quantifying immune subsets and markers on the primary tissue at surgery to exclude the possibility that increased DFS could be attributed to greater immunogenicity of the tumor at baseline. We have added here the analysis of baseline absolute lymphocyte count with DFS where we observed no significance as new Supplemental Figure 2k as follows:

This is also referenced in the text on lines 169-172 as follows:

“There was no correlation between DFS and lymph node status, primary tumor size, baseline CA19-9 levels, ctDNA positivity at baseline, neoadjuvant chemotherapy, splenectomy, or absolute lymphocyte counts at baseline (Supplemental Figure 2e-k).”

3. The vaccine fails to consistently generate robust T cell responses to KRAS G12D, the most common mutation in PDAC and colorectal cancer. Other studies have reported robust responses against these epitopes. G12D and G13D antigens are highly common and pose important clinical relevance, furthermore, weaker response to G12D and G13D is concerning and undercuts the rationale of a pooled vaccine approach if key subtypes remain poorly immunogenic. Accordingly, the authors need to address such finding.

Response: We agree that G12D is the most prevalent in PDAC, and this study is one of the first to report that KRAS G12D induces the lowest T cell responses in the context of vaccination. To our knowledge, prior studies testing mKRAS vaccines have not directly compared the relative peripheral T cell immunogenicity across all six of the common mKRAS antigens and are thus not able to provide direct evidence of this. We agree, however, this brings to light important considerations for a pooled vaccination approach, and reason that the cross-reactive T cells across antigens may support such a pooled approach. In this case, vaccination with other antigens may stimulate cross-reactive responses to the G12D antigen. We have acknowledged this point and expanded on this on in the discussion on lines 441-462 as follows:

“Little is known about the immunogenicity ranking of antigens included in vaccines, particularly when administered as a pooled vaccine. Here, we are the first to test a pooled vaccine against 6 common KRAS mutations. Interestingly, we observed weaker immunogenicity of mKRAS G12D independent of whether the patient’s tumor expressed this mutation. In contrast, T cell responses to mKRAS G12V and G12R were of a higher magnitude than mKRAS G12D and G13D, findings that are consistent with preclinical models²⁷. As KRAS G12D is the most common mutation detected in PDAC tumors (~40%), this highlights the need to improve immunological responses against this mutation. One approach may be vaccination with a single tumor-specific peptide vaccine in the therapeutic setting, which could be explored in future studies. However, in the preventative vaccination setting where a potential KRAS mutation is unknown, single peptide vaccination limits immunological coverage across mutations. Additionally, another question remains if the cross-reactive T cells raised by the pooled vaccine in this cohort may also be advantageous for providing broader immunological protection across antigens. The low immunogenicity of mKRAS G12D also coincides with decreased DFS, suggesting that different mutations may have different rates and/or mechanisms of tumor growth and TME immunosuppression^{28,29}. Reduced OS has been shown for mKRAS G12D positive tumors compared with G12R tumors, but not relative to G12V tumors. However, in our *albeit* 12 patient cohort, we observed a significant improvement in DFS for patients with G12V mutations relative to G12D positive patients³⁰. Larger studies are required to further examine the clinical significance of this preliminary observation.”

4. The significance of identifying “public” TCR clonotypes is an intriguing finding but not yet clinically actionable and there is insufficient discussion on how this information can guide patient selection, future vaccine design, or TCR-engineered therapy etc.

Response: We agree that this is one of the first reports of public KRAS TCR clonotypes and thus provides important novel biological information to the field in regard to our understanding of the basic immunology of responses to these antigens as well as defining targetable patient populations for antigen-specific vaccines. We have expanded on the significance of these TCRs in greater detail in the Discussion on lines 511-525 as follows:

“Importantly, we also identified public mKRAS TCR β clonotypes within our dataset. One of these mKRAS TCRs identified in two patients on our trial was also detected in the tumor infiltrating lymphocyte population of a patient by Levin *et al.*²⁴, further supporting the detection of public clonotypes across patients outside of our cohort. The alpha chains of the public TCR β 06 found in

a previous study and HS-public TCR β 08 found in our study were different; however, both clonotypes responded to KRAS G13D indicating a potential TCR β chain-dominant interaction with cognate peptide-HLA²⁴. We confirmed TCR β 08 had conserved reactivity to the G13D antigen in the context of DQB*03:01 identifying that patients with this mutation-HLA combination may be an ideal population for vaccination with this antigen in future clinical settings. Additionally, these public clonotypes may be ideal candidates for adoptive TCR therapies based on their conserved immunological relevance across patients. While public TCR clonotypes have been identified against many pathogens⁴⁷, little is known about the abundance and role of public or HS-public clonotypes against cancer antigens. Thus, this work provides important basic immunological context of shared T cell responses to these highly conserved neoantigens across human solid tumors. To our knowledge, this is the first report of public mKRAS-specific TCR clonotype detected in patients vaccinated with any mKRAS vaccine.”

5. There is minimal evidence that the identified TCRs actually recognize endogenous tumor antigen in physiologic settings, mediating the direct anti-tumor activity in vivo. The data reported in the manuscript demonstrated only weak responses in the form of antigen recognition and T cell activation were seen in assays with Jurkat and cancer cells, largely reliant on exogenous peptide pulsing (peptide pulsing is artificial and may overestimate functional relevance).

Response: Given the predominance of CD4 T cell response, we prioritized reactivity validation of CD4 TCRs. We aimed to validate our novel findings in regard to the TCR biology by including: 1) mKRAS reactive TCRs that were mono- and cross-reactive to antigens, and 2) identified public TCRs that could respond to antigen in all of the patients in whom the TCR was identified. While we did not find these CD4 T cells directly recognized the tumor cell lines tested, this may demonstrate their potential role in indirect tumor cell clearance that has been reported in several clinical and preclinical studies (Alspach *et al.* Nature 2019, Kruse *et al.* Nature 2023, and Mumberg *et al.* PNAS 1999). To support this, we have added experimental data showing TCR05 was able to recognize patient matched LCLs when pulsed with tumor cell lysate which expresses cognate KRAS mutation antigen as new Supplemental Figure 9F:

And has been referred to in the results on lines 404-409 as follows:

“To test if TCR05 can indirectly recognize target antigen, we pulsed patient matched LCLs with tumor cell lysate generated from control or CAPAN1 (G12V+) tumor cells (**Supplemental Figure 9f**). Indeed, TCR05 expressing human T cells increased CD25 expression after co-culture with CAPAN1 but not control lysate pulsed APCs supporting a role for indirect recognition of tumor with these identified mKRAS-specific TCRs.”

6. The dominant vaccine-induced response reported by the authors is CD4 T cell based, with CD8 Teff cell population detected only in a limited number of patients and was most likely exhausted. Yet, effective tumor clearance typically relies on cytotoxic CD8+ T cell activity, especially in poorly MHC-II expressing tumors like PDAC. The manuscript offers no compelling evidence or reasons for these findings and also does not offer any strategies that may be used to overcome this while designing future vaccines.

Response: While the proportion of responding CD8 T cells was lower than responding CD4s, our predominant responding CD8 population, CD8 Tem Acv, does not have markers of terminal exhaustion (Figure 3A) as shown here:

Figure 3A:

Figure 3D:

The CD8 Tem Acv cell population which is increased across patients to most antigens (Figure 3D, left) express high levels of CD137 activation marker, functional cytolytic cytokine GzmB, and are highly proliferative by KI67 positivity. They express moderate levels of PD1, consistent with canonical markers of contraction after activation, but do not express high levels of CTLA4, TIM3, or LAG3. The second activated CD8 population observed, CD8 Teff Acv, does have higher expression of dysfunctional inhibitory receptors but was a minor responding population across all patients or antigens and was not a significant response across the cohort (Figure 3D, right panel).

Regarding the physiological role of responding CD4 T cells, several studies have shown that CD4 T cells can mediate tumor cell clearance either through direct or indirect mechanisms. Additionally, we demonstrate that 9/12 patients in our cohort have MHCII+ tumor at baseline (Supplemental Figure 3A) – thus, providing a feasible mechanism for tumor recognition of some CD4 clonotypes raised by the vaccine. Additionally, we have added experimental evidence these T cells can function through indirect antigen recognition as shown in reviewer point 5. We have clarified this in our discussion on lines 464-493 as follows:

“Cancer vaccines must not only generate a diverse activated T cell response for tumor killing but also elicit memory T cells should recurrence occur. We show, through single cell proteomics and transcriptomics, that mKRAS-VAX activates mKRAS-specific Th1 CD4 central and effector memory T cells that express functional cytokines IL2, IFN γ and TNF α as well as cytotoxic effector memory CD8 T cells which express the cytotoxic effector molecules GzmB. The Th1 CD4 responses observed are consistent with other peptide-based vaccine approaches and have previously been associated with improved clinical outcomes³¹. Notably, and consistent with our approach, the amphiphile-peptide mKRAS vaccine also elicited predominantly Th1 CD4 T cells, while in contrast no mKRAS-specific T cells were detected by mRNA-based vaccine¹⁹. Additionally, the mRNA-based approach mounted predominantly CD8 T cells to the targeted personalized neoantigens, highlighting the need for more study of the differences of T cell populations activated by each platform.²² Lower levels of the Th2-associated cytokine IL-5, but not IL-4, were also detected after mKRAS peptide stimulation in this study. While IL-5 function is still not entirely understood in the context of immunotherapy, recently, IL-5 secreting CD4 T cells have been associated with efficacy of ICB by promoting eosinophil expansion, tumor infiltration, and thus intra-tumoral CD8 T cell activation in breast cancer³². In our study, we found that the activated T cell populations persisted and tracked with a durability marker CCR5 in the four patients who remained enrolled during the extended vaccine-only, no-ICI phase. The CD4 Tem population was the most durable, likely owing to the trafficking of CD4 Tcm to the secondary lymphoid tissues over time due to CCR7 expression. Furthermore, granzyme B was detected in responding mKRAS-specific CD4 T cells, questioning the potential for a role of CD4 T cells in direct tumor killing. In the small subset of TCRs validated in our study, T cells recognized antigen when peptide was supplied exogenously or when KRAS-mutation specific tumor cell lysate was pulsed onto APCs, suggesting their function in controlling tumor growth may be more related to indirect mechanisms of tumor control as previously reported³³⁻³⁷. Notably, Alspach *et al* described the critical role of neoantigen-specific CD4 T cell-mediated control of tumor growth in tumors that lacked expression of MHCII, underscoring the potential importance of these T cell responses³³⁻³⁷. We also found that four patients who remained recurrence free displayed moderate levels of HLA-

DR, DP and DQ in their primary tumor tissues, again suggesting that CD4 T cells may directly recognize patient tumors. However, additional testing of mKRAS-specific CD4 T cells in our dataset and their ability to lyse tumor directly needs to be explored. Nonetheless, these data appear to be consistent with high levels of HLA class II associated with greater activation of T regulatory cells and a more immunosuppressive TME³⁸.”

7. The study largely depends on surrogate markers (ELISPOT, cytokine profiling, CyTOF) and lacks functional anti-tumor assays. Direct tumor lysis by vaccine-induced T cells was not shown, especially for CD4+ T cells, which as mentioned above was the major response induced by the vaccine; CD4 T cells alone are not sufficient for tumor clearance or sustained control.

Response: Please refer to response to point 5.

8. Results in multiple instances are either missing or are inadequately described. This include but not limited to:

a. Suppl Fig. 8F: GSEA pathway analysis data is missing.

Response: We apologize for this omission. We have now added the data for our KEGG pathway analysis performed on this dataset as new Supplemental Data 3 and have clarified this in the text on line 320 as follows:

“At a gene level, we observed changes broadly in genes associated with metabolism, cytoskeletal structure, and inflammation in the post-vaccine CD4 Tcm and CD8 Tem populations, although we did not observe any significant changes at the pathway level by KEGG pathway analysis (**Supplemental Figure 8f, Supplemental Data 3**).”

b. Figure 6c: Naive, proliferating, TEM CD4 T cells are also shown but are not described in the results in the text.

Response: We thank the reviewer for this point. The mKRAS-specific T cells mapped onto the single cell dataset aligned predominantly to the CD4 Tcms and therefore focused on this population. In the other populations (CD4 naive, proliferating, TEM), there were not enough cells to perform this statistical analysis with confidence in our phenotype analysis. We clarified this in the manuscript on lines 338-340 as follows:

“We compared the gene expression profiles of mKRAS-specific CD4 Tcm cells relative to all other CD4 Tcm cells in the dataset. We note that too few mKRAS-specific cells were detectable in the CD4 naïve, proliferating, or Tem populations to perform this analysis (**Figure 6c, Supplemental Table 8**).”

c. Lines 349-351, Fig. 6e: Authors mention- TCR03, a cross-reactive TCR that had expanded in vitro to KRAS G12V, G12A, and G12C responded significantly to these three

antigens at the highest peptide concentration tested (10 microM). However, data at different concentrations is not shown (similar to what is shown in 6d).

Response: The response to antigen by TCR03 was observed to be low relative to others TCRs when stimulated at the highest concentration. TCR03 did not respond at lower concentrations, thus we were unable to make an avidity curve similar to those in 6d. We have instead highlighted its weak response at high concentrations for clarity on lines 362-364 as follows:

“TCR03, a cross-reactive TCR that had expanded *in vitro* to KRAS G12V, G12A, and G12C responded significantly to these three antigens at the highest peptide concentration tested (10 μM) but did not respond at lower concentrations of peptide.”

d. Fig. 6i: Please explain why T cells were also activated in SUDHL10 cell line when exogenous mKRAS peptides were provided.

Response: SUDHL10 cells express WT KRAS and also express HLA-DRB*07:01, the cognate HLA for the TCRs tested, and thus serve as antigen negative controls in the absence of peptide in our co-culture system. We supplied exogenous mutant antigen to demonstrate that, when the negative control which does express the correct HLA was supplied with target antigen, this allowed recognition by cognate T cells further confirming specificity of the TCR. We have clarified that in the results section on lines 403 as follows:

“However, strong activation was detected when each mKRAS antigen was supplied exogenously to either CAPAN1 or SUDHL10 cells, which expresses the cognate HLA allele. (Figure 6h).”

Minor comments:

Response: We thank the reviewer for these minor but important comments and have amended these throughout the text.

1. All figure numbers should be referred to in lower case both in the text and figure legends (same as in the figures).
2. The figures are not referenced correctly in several instances:
 - a. Line 207: Fig. 3c should be 3a.
 - b. Line 224: Refer Fig. 3a and Fig. 3d.
 - c. Line 225: Correct to Suppl. Fig. 6a-e.
 - d. Line 241: Fig. 4f should be 4e. Also correct this labeling in the figure and figure legend.
 - e. Line 319: Suppl. Fig. 8h is not referred anywhere. Referred to on line 334.
 - f. Suppl Fig. 9c: In the figure legend, b should be c and c should be b.
 - g. Figure 6: 6g is mentioned two times in the figure legend (should be changed to 6e). Also, 6f is not included in the figure legend. Therefore, it is not clear where TCR05 data is shown as mentioned in the text on lines 355 to 357.
 - h. Line 361: Suppl Fig. 8d should be changed to Suppl Fig. 7g.
3. Line 238: timepoint should be post 52 weeks as mentioned in the figure legend.

4. Please proofread as several sentences are repeated or have some other errors, such as:
 - a. Line 248: Despite the predominance of the mKRAS oncogene in of PDAC.....Remove "of".
 - b. Line 261: Add- "in patients who recurred" at the end of the sentencegreater clonality of the tumor-specific.....
 - c. Line 265: Insert "in" towere not present the pre-vaccine.....
 - d. Line 285: space between oftentimes.
 - e. Line 309: Remove: "in".
 - f. Fig. 5b legend: Replace "and" with "at".
 - g. Suppl. Fig. 9e: Correct figure legend- T cells from a healthy donor "were" (not with) isolated. Also correct: first endogenous TCR was KO and then TCR5 was KI (it is written the other way round); same correction needed in Fig. 6i legend. This order is correct. We knock in the transgenic TCR of interest, then knock out endogenous. Our transgenic TCR construct has a codon optimized PAM site in the TRAC/TRBC locus that prevents editing of the transduced TCR construct allowing for this order which gives us optimal transduction. This was clarified in the Methods.
 - h. Fig. 6b legend: Insert space between conditions mapped.
 - i. Fig. 6i legend: TCR06 should be TCR05.
 - j. Fig. 6h legend: MDA-MD-231 should be MDA-MB-231.
 - k. Lines 421-422: Repeated.
 - l. Lines 425-426: Repeated.
 - m. Lines 421-422: Repeated.
 - n. Line 534: Correct to: spot forming unit.
 - o. Lines 547-550: Repeated.
 - p. Line 574: Correct SUDL10.
 - q. Lines 679-682: Repeated.
 - r. Lines 722-723: Repeated.
 - s. Suppl. Fig. 3a legend: Remove extra 'and'. The 'and' in Supp Fig 3a refers to the CD4 and CD8 costain. We clarified this by indicating each other stain was a single stain in the figure legend.
 - t. Suppl. Fig. 4a legend: Remove extra 'including'.
 - u. Suppl. Fig. 8f legend: Replace T cell populations with "in".
 - v. Suppl. Fig. 8g legend: Insert "(").
5. Fig. 2a legend: provide info about immune responders and non-responders.
6. Lines 318 to 321: Include the names of genes.
7. Figure 6b: Do the colors of the bars match with the legend of 6a? Please provide the legend in 6b or clarify.
8. Lines 327-332: The 5th gene, APOBEC3H is not mentioned.
9. Line 342: Correct that Jurkat cells were transduced with mKRAS-reactive TCRs (not T cells).
10. Suppl Fig. 9b: It would be good to write what cherry+ represent in the figure legend and also on the Y-axis in Fig. 9b.
11. Suppl Fig. 9d: Also describe results of MDA-MB-231 cells with G13D KRAS as shown in SF9d in the text.

Reviewer #3

(Remarks to the Author)

The manuscript by Huff et al. is a comprehensive analysis of a Phase 1 study assessing the safety and immunogenicity following vaccination with a pool of synthetic long peptides targeting 6 KRAS subtypes administered with Poly ICLC in patients with resected pancreatic ductal adenocarcinoma. Overall, the trial is well-performed by experienced and outstanding investigators focused on immune therapies against PDAC. The paper demonstrates safety of the vaccine regimen in the context of combination immune-checkpoint blockade. The major strength of the paper is the detailed analysis of the immune responses over time using a variety of assays. As the trial was done in 12 patients, it was not powered to establish immune correlates with clinical outcome although there is an exploratory analysis of this.

Summary of key results

1. CD4 responses were generated in 11/12 patients. CD8 responses were low to undetectable. The authors need to highlight the limiting CD8 T cell responses in the discussion. Notwithstanding the data showing CD4 T cells can have an important protective role against tumors, the data presented highlight that this vaccine approach would be limited for CD8 T cell generation.
2. Responses to mKRAS G12V were highest and G12D and G13D the lowest
3. T cell cross reactivity against multiple KRAS antigens were generated.
4. A public KRAS-specific TCR clonotype was discovered following vaccination.

Originality and significance

The paper uses a set of mKRAS peptides for vaccination which is likely new. The use of Poly ICLC as an adjuvant is not novel. The immune analysis is extensive although recent work by Balachandran for personalized neoantigen cancer vaccines against PDAC is equally if not more detailed. The TCR analysis is more extensive than what has been done but not novel.

Data and methodology: The techniques used (Cytof, elispots, flow cytometry and rRNAseq are valid.

Statistics- Seems valid- using fold change for a lot of their analysis can be less informative than seeing the actual data.

Conclusions- Generally supported by the data.

Clarity- Some editing needs to be done. The manuscript is dense and there are areas for improving clarity.

Major comments:

1. vaccines were administered subcutaneously at 5 injection sites. Can the authors verify that all 6 peptides were pooled and then given at the 5 different sites. Have prior studies been done to optimize the dose of Poly ICLC and the regimen (single peptide per site versus combining all peptides) for how to achieve optimal responses. The dose of 5ug of Poly ICLC seems relatively low compared to what is used in pre-clinical studies.

Response: We confirm that all six peptides were pooled and given at 5 different sites. Given the hydrophobic nature of the peptides, we had to dilute our peptides to a volume that required us to deliver the vaccine in 5 injections.

The Poly ICLC dose is a typo, and we will correct the dose to be 0.5mg. This same dose has been used in a number of human vaccination trials with synthetic long peptides (including Ott *et al*, Nature 2017, Ott *et al.*, Cell 2020). This dosage and vaccination schedule has been previously established to elicit immune responses in humans.

We have clarified our vaccination strategy in our Methods section on in the Vaccine formulation and administration section starting on line 569 and as follows:

“The study vaccine consists of 21-mer synthetic long peptides corresponding to six of the most common KRAS mutation subtypes found in PDAC. Peptides were synthesized by JPT Technologies with sequences as follows: G12V- YKLVVVGAVGVGKSALTIQLI, G12C- YKLVVVGACGVGKSALTIQLI, G12A- YKLVVVGAAGVGKSALTIQLI, G12R- YKLVVVGARGVGKSALTIQLI, G12D- YKLVVVGADGVGKSALTIQLI, G13D- YKLVVVGAGDVGKSALTIQLI. These six peptides were admixed (0.3mg/peptide, a total of 1.8mg) with 0.5mg TLR3 agonist poly-ICLC (Hiltonol; Oncovir) in saline prior to vaccine administration at the Investigational Drug Service Pharmacy at Johns Hopkins Hospital. The pooled vaccine consisting of all six peptides was administered subcutaneously at five injection sites.”

2. While the study was not designed to correlate DFS with immune analysis, the authors stratify the patients into two groups above and below the 25 percentiles of the maximal fold-change or average tumor specific T cells. Using fold-change may one way to present complex data, but it preferable to show that actual data for elispot responses across the different peptides and denote whether they progressed. In Figure 2b, the symbols could be changed to those that had DFS. This would more clearly highlight whether the 4 highest responders to G12V did not progress. The elispot response are generally between 10-50 for each of the mKRAS peptides which relatively low compared to viral vectors or prime-boost approaches. This may be a concern for durable immunity/protection.

Response: We thank the reviewer for these points. While we agree the absolute magnitude was observed to be relatively lower for some antigens in some patients, there have been many instances of lower detectable IFN γ responses *ex vivo* with clinical outcome. For example, in Ott *et al*. 2017, the two patients with clinical responses after personalized neoantigen vaccination in their cohort had responses to antigens with less than 100SFU, while clinical non-responders had higher immune responses overall (Ott, P *et al*, Nature 2017 - Figure 2B). Analysis of tissue specimens after vaccination may elucidate neoantigen-specific T cell trafficking and possibly account for these differences- which our studied was limited by. We have added this point to our discussion of the limitations of our study analysis on lines 428-439 of the discussion as follows: “

“Patients were allowed to enroll within 6 months of completing adjuvant chemotherapy and therefore joined the study at different timepoints within this window. Our exploratory analysis suggests that patients in the top three quartiles of average tumor-specific T cell responses (by ELISPOT) experienced greater DFS, which we have defined from the time of first vaccination. Notably, we did not observe a correlation of DFS with patient characteristics, tumor size, lymph node status at resection, CA-19-9 levels, ctDNA positivity, neoadjuvant chemotherapy, splenectomy, or lymphocyte count at baseline. Although perhaps informative for a potential magnitude of response that is required for clinical benefit, these correlations should be interpreted with caution given that the trial was not powered for clinical efficacy but appears to be consistent with recent neoantigen vaccine studies in PDAC^{19,22}. Our study was also limited in analysis of post-vaccine tumor samples which were unavailable in this cohort. Future studies which correlate mKRAS-specific T cell infiltration into tumors with clinical outcomes may also be more informative.”

Additionally, to address this, we have added symbols in Figure 2D that indicate patients who recurred vs those who have not recurred as follows:

New Figure 2d:

3. The authors use CyTof and Flow cytometry to characterize the phenotypic and functional cytokine responses. This is used to delineate the responses into various CD4 and CD8 effector and memory populations. It would be clearer if they used a standard set of markers to define the subsets. CD45RA and CD45RO is a more general way to divide the T cell populations but lacks greater depth of the delineating the populations into distinct phenotypic subsets. As they have CCR7 and CD62L in their analysis, they can use these markers to better delineate the populations of the cells. The authors need to define what markers were used to define each of the T cell subsets in Figure 3.

Response: This is an important point, and we have added expression of CCR7 into the phenotyping heatmap used in new Figure 3A below. We apologize for this initial oversight.

While we had CD62L on our flow cytometry panel, we did not have this marker on our cytof panel. We have better defined in the text the markers that were used to define each cell type in the results section on lines 202-204 and lines 219-225 as follows:

“Within the CD4 compartment, we noted a significant increase in activated mKRAS-specific CD4 central memory (Tcm, CD45RO+CD45RA+CCR7+) and effector memory (Tem, CD45RO+, CD45RA-, CCR7-) populations at the peak and late timepoints relative to baseline (**Figure 3c**).”

And

“Within the CD8 compartment, the activated effector population (Teff Acv, CD45ROhi, CD28hi) expressed highest levels of the activation markers CD137, OX40, KI67, cytokines IFN γ , IL-2, and granzyme B, as well as the exhaustion markers CTLA4, TIM3, and PD1..... Conversely, a significant increase in activated CD8 effector memory cells (Tem Acv, CD45RAlo, CD45ROlo, CCR7-) was detected post-vaccine for G12V, G12A, G12R, G12C, and G13D across patients.”

New Figure 3a:

4. Line 205-206 says that Tem Acv cells express the highest level of cytokines (IL-2, TNF, IFN). This is not true for IFN.

Response: We apologize for this inconsistency and have corrected this statement in the text on line 209.

5. It is notable and somewhat surprising that the CD4 T cells produce IL-5. Also, the IL-10 is detected in the stimulated cultures (Supplemental Fig 5) which was not mentioned in the text. While the significance of these Th2 and regulatory cytokines are not clear, there should be mention of this in the discussion (line 435) when discussing the responses generated.

Response: We agree this was an interesting and unexpected observation. We have added a discussion of this observation on lines 474-478 as follows:

“Lower levels of the Th2-associated cytokine IL-5, but not IL-4, were also detected after mKRAS peptide stimulation in this study. While IL-5 function is still not entirely understood in the context of immunotherapy, recently, IL-5 secreting CD4 T cells have been associated with efficacy of ICB by promoting eosinophil expansion, tumor infiltration, and thus intra-tumoral CD8 T cell activation in breast cancer³²”

6. In Figure 4b, the authors present data as a normalization to the control following 48-hour stimulation with peptide. Do the authors have a kinetic of the immune responses with the peptides. A 48hour stimulation with peptides is longer than what is standard for the field in measuring responses and in other models, a long stimulation can lead bystander enhancement.

Response: Initial peptide stimulation kinetic studies performed did indicate that 48-hours was the optimal time point for detecting responding mKRAS-specific T cells. Please see below for our initial characterization of cytokine expression from total T cell populations in Response Figure 1. As a similar comparison for detecting vaccine induced mKRAS-specific T cell responses, Pant, S. et al Nat Med 2024, stimulated post-vaccine PBMCs for 44h, consistent with our methods.

Response Figure 1:

7. It seems counterintuitive that the only durable detectable responses are Tem Acv rather than Tcm Acv.

Response: We agree this was an interesting finding. We reasoned that over time, the Tcm population likely homed to lymphatic tissue due to the observed expression of CCR7 (by CyTOF, Figure 3a) and therefore the remaining mKRAS-specific T cells in circulation at later time points were more Tem. We have pointed this out in this discussion on lines 480-481 as follows:

“The CD4 Tem population was the most durable, likely owing to the trafficking of CD4 Tcm to the secondary lymphoid tissues over time due to CCR7 expression.”

8. Line 323- Should this refer to Supplemental Figure 8E and if so, the figure shows that CD4 responses are higher in two of the patients, while CD4 and CD8 responses are similar in the other two.

Response: We thank the reviewer for identifying this discrepancy. This call out should refer to Supplemental Figure 8H and was corrected in the new manuscript file on line 334 as follows:

“We noted that most CD8 T cell TCRs found in the adaptive sequencing datasets were present at a lower frequency (<0.1% of repertoire) compared to CD4 T cells; this is consistent with a predominant CD4 T cell response to our vaccine (**Supplemental Figure 8h**).”

9. The data in Figure 6c seems to break the flow. Perhaps the more interesting/important data should be the TCR analysis and the differences in a small subset of genes be taken out or included in some of the supplemental data.

Response: As suggested, we have moved this figure to the supplemental section and is now new Supplemental Figure 8i.

10. While the TCR analysis is interesting and having additional KRAS specific TCR sequences may be important in using for T cell therapy, the functional data they show for G12V required exogenous peptide with limited reactivity against the endogenous KRAS G12V suggests that they may limited in functionality in vivo.

Response: To further support the potential *in vivo* function of the validated T cells outside of direct tumor recognition, we have provided new experimental data that demonstrate recognition of antigen from APCs pulsed with whole tumor cell lysate- please see reviewer 2, point 5 above.

11. The discussion should be edited. From lines 432-449 the authors seem to make a case for CD4 T cells in having a protective role. There is a lot of back and forth in this section. I cannot understand how the data in the paper are consistent with greater activation of T regulatory cells and a more suppressive TME.

Response: We apologize for the confusion in this paragraph. We aimed to highlight the identification of the mKRAS-specific CD4 T cells which had a Th1-like phenotype and expressed effector cytokines such as IFN γ , IL2, and TNF α which may promote a pro-

immune/anti-tumor tumor microenvironment. We have clarified this paragraph in the discussion on lines 464-470 as follows:

“Cancer vaccines must not only generate a diverse activated T cell response for tumor killing but also elicit memory T cells should recurrence occur. We show, through single cell proteomics and transcriptomics, that mKRAS-VAX activates mKRAS-specific Th1 CD4 central and effector memory T cells that express functional cytokines IL2, IFN γ and TNF α as well as cytotoxic effector memory CD8 T cells which express the cytotoxic effector molecules GzmB. The Th1 CD4 responses observed here are consistent with other peptide-based vaccine approaches and have previously been associated with improved clinical outcomes³¹.”

12. The authors may want to include the reference by Bear et al. JCI 2024 based on the identification of TCRs targeting KRAS G12V.

Response: We thank the reviewer for pointing this out. While we cited earlier literature from Bear *et al.*, we missed adding this newer study. We have added this reference on line 495.